# DISC: Dynamic Decomposition Improves LLM Inference Scaling

**Jonathan Light**[1,2,4][*†], **Wei Cheng**[2][✉], **Benjamin Riviere**[4], **Yue Wu**[3], **Masafumi Oyamada**[5],
**Mengdi Wang**[3], **Yisong Yue**[4], **Santiago Paternain**[1], **Haifeng Chen**[2]
[1]Rensselaer Polytechnic Institute, [2]NEC Laboratories America, [3]Princeton University,
[4]California Institute of Technology, [5]NEC Corporation

[disc-search.github.io](disc-search.github.io)

## Abstract

Inference scaling methods for LLMs often rely on decomposing problems into steps (or groups of tokens), followed by sampling and selecting the best next steps. However, these steps and their sizes are often predetermined or manually designed based on domain knowledge. We propose dynamic decomposition, a method that adaptively and automatically partitions solution and reasoning traces into manageable steps during inference. By more effectively allocating compute – particularly through subdividing challenging steps and prioritizing their sampling – dynamic decomposition significantly improves inference efficiency. Experiments on benchmarks such as APPS, MATH, and LiveCodeBench demonstrate that dynamic decomposition outperforms static approaches, including token-level, sentence-level, and single-step decompositions, reducing the pass@10 error rate by 5.0%, 6.7%, and 10.5% respectively. These findings highlight the potential of dynamic decomposition to improve a wide range of inference scaling techniques.

## 1 Introduction

Scaling inference efficiency remains a fundamental challenge for large language models (LLMs). Many existing approaches improve inference by decomposing problems into smaller steps and systematically exploring different solutions [1, 2, 3, 4, 5, 6, 7, 8, 9, 10]. Some decomposition methods often rely on domain-specific heuristics and hand-crafted rules [11, 12, 13]. However, manually partitioning problems or designing task-specific heuristics is costly and lacks generalization. Moreover, identifying critical steps for an LLM can be non-trivial for humans. LLMs may assign importance to seemingly trivial words (e.g., therefore or which), which, while counterintuitive to humans, play a crucial role in autoregressive generation [14]. Other approaches employ fixed, uniform step sizes, such as token- or sentence-level decomposition [1, 15]. All these methods rely on **static decomposition strategies**, where step sizes are predefined or determined via heuristics. Such rigidity risks overusing compute on steps that are easy for the LLM (but potentially difficult for humans) while undersampling more challenging steps.

To overcome these limitations, we propose DISC (Dynamic decomposition Improves Scaling Compute), a recursive inference algorithm that dynamically partitions solution steps based on difficulty. Unlike prior methods, DISC **adapts decomposition granularity** during inference based on both the available budget and problem complexity, ensuring finer granularity for more difficult steps. By leveraging the autoregressive nature of LLMs, DISC efficiently **locates difficult steps** through dynamically proposing step sizes, focusing compute on challenging regions rather than wasting

---

[*]Work done during the part-time internship at NEC Laboratories America. [✉]Corresponding author.
[†]This work was partially supported by IBM through the IBM-Rensselaer Future of Computing Research Collaboration.

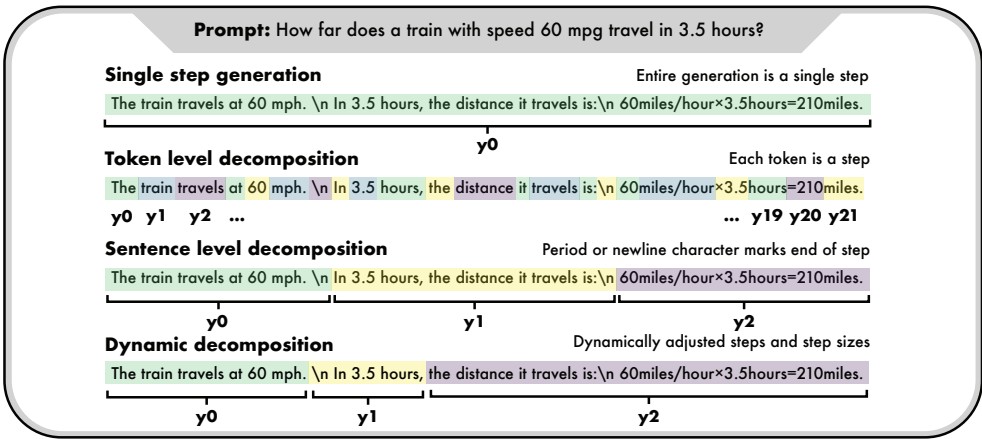

Figure 1: **Comparison of automatic decomposition strategies based on step size.** Coarser steps accelerate the search process but risk skipping over optimal solutions and committing to suboptimal prefixes. In contrast, finer steps ensure more precise decisions but lead to slower search. A dynamic strategy that adapts step size based on LLM feedback offers a balanced approach, combining the efficiency of coarse steps with the precision of fine-grained decomposition.

resources on trivial steps. DISC is generalizable, requires *no human supervision, domain-specific heuristics, prompt engineering, or process annotations*, and is easily *parallelizable*, making it widely applicable across tasks. Furthermore, DISC is plug-and-play with off-the-shelf search algorithms and can be naturally integrated with greedy, beam, or Monte Carlo Tree Search.

Our main contributions are:

- We introduce **DISC**, a method for dynamically adjusting step sizes and decomposing solutions during inference without human supervision, domain-specific heuristics, or process reward models.
- We demonstrate how DISC integrates decomposition directly into inference-time search, **allocating compute more effectively toward high-potential solution prefixes**.
- We show that DISC improves inference scaling in terms of both **sample efficiency**, **token efficiency**, and **runtime**, achieving up to **10% reduction** in error relative to the baselines and up to **4x increase** in accuracy over the base model, with just 10 samples, including with reasoning models.
- We provide both empirical and theoretical insights into LLM reasoning, including identifying **critical intermediate steps** and analysis of how DISC helps discover optimal solutions.

## 2 Preliminaries

### 2.1 Problem Setting

We consider a reasoning and code generation setting where we are given: a dataset $\mathcal{X} = \{\boldsymbol{x}^{(i)}\}_{i=1}^{N}$, a pretrained, autoregressive LLM $\pi$ that generates solutions $\boldsymbol{y} \in \mathcal{Y}$, and a reward model $R :$ $\mathcal{X} \cdot \mathcal{Y} \to [0, 1]$ that evaluates the generated solutions. The goal is to find inputs to the LLM that produce solutions with high reward. This setting includes program synthesis, where correctness is verified using ground-truth tests [16, 17], and mathematical reasoning, where solutions are validated numerically [18, 19]. The reward model can be a ground-truth verifier, a trained heuristic [20], self-consistency [21], or an LLM-as-a-judge [22] and because our focus is on decomposition rather than verification, we use the ground-truth reward model where available.

We use the following hierarchical token notation: $y$ is a token, $\boldsymbol{p}$ is a prefix that starts at the prompt and concatenates multiple steps, $\boldsymbol{s}$ is a suffix that concatenates multiple steps and ends with the EOS token, and $\boldsymbol{y}$ is a complete solution that starts at the prompt, concatenates multiple steps, and ends at the end of sequence token EOS. We denote a concatenation of two tokens or token sequences with $\cdot$, e.g. $\boldsymbol{p} \cdot \boldsymbol{s}$ is the concatenation of prefix $\boldsymbol{p}$ and suffix $\boldsymbol{s}$ to form a complete solution $\boldsymbol{y}$. We denote the sampled suffix from prefix $\boldsymbol{p}$ using the LLM policy as: $\boldsymbol{s} \sim \pi(\cdot|\boldsymbol{p})$. We denote an optimal solution with $\boldsymbol{y}^*$ and an optimal suffix as $\boldsymbol{s}^*$, where there exist multiple optimal solutions. For our analysis,

we use the convention that $\pi(s^*|p) = 0$ if there does not exist a completion $s^*$ such that $p \cdot s^*$ is optimal. The **size** of a string $|y|$, refers to its length in tokens or characters.

## 2.2 Existing Decomposition Methods

**Single-step generation.** In a single-step generation, the entire solution is generated in one pass from the prompt to the EOS token, treating it as a single action. This approach underlies the widely used inference scaling method **best of n** (BoN) [19, 23, 5, 24], where $n$ complete solutions are sampled, and the highest-scoring one is selected. Single-step generation also plays a role in alignment and fine-tuning methods such as **DPO** [25] and **RLOO** [26].

**Token-level decomposition.** At the opposite end of the spectrum, token-level decomposition treats each atomic token as an individual step. While this approach dramatically increases search complexity, it enables fine-grained search that can yield higher performance gains given sufficient compute [1].

**Newline and sentence-level decomposition.** A commonly used decomposition method segments LLM generations into sentences or lines based on delimiters such as periods or newlines [27, 1, 11]. Typically, each newline corresponds to a new paragraph, equation, or line of code, which often encapsulates a distinct reasoning step.

> **Challenge: Automatic and scalable decomposition**
>
> Existing decomposition methods are static and manually designed, resulting in either slow convergence to good performance or fast convergence to poor performance. We propose adaptive decomposition for fast convergence to a good performance.

## 2.3 Search for Inference Scaling

We differentiate between decomposition and search, where decomposition controls the number of steps between new branches and search is the process of selecting which branch to explore. In other words, decomposition is the construction of nodes and edges, and search is a process that occurs on that structure. In this work, we propose a decomposition method that is plug-and-play with different search methods. In our experimental results in Sec. 4.1, we compare decomposition methods: token-level decomposition, sentence-level decomposition, and DISC with the search methods: greedy, beam [28], and Monte Carlo Tree Search [1, 29]. Implementation details are provided in App. F.

# 3 Methodology

The DISC algorithm uses LLM rollout data to dynamically decompose reasoning steps, adjust step sizes, and allocate sampling compute. In Sec. 3.2, we present DISC paired with a greedy search strategy, shown in Alg. 1 and Fig. 3. In Sec. 3.4, we present the general case of DISC as a nodal expansion operator which is used for pairing with beam search and MCTS.

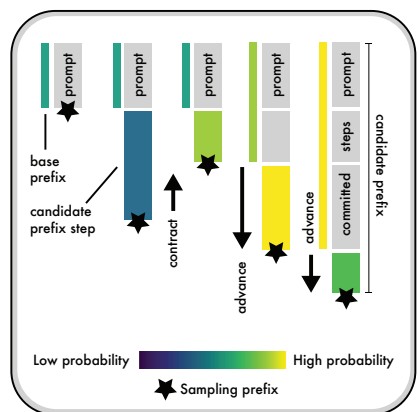

## 3.1 High-level Overview

At a high level, DISC with Greedy Search advances a base prefix $p_b$ forward by iteratively concatenating promising steps to it. The core intuition is to dynamically allocate compute by adjusting the step size: if a prefix shows promising reward improvement, we take a large step forward; if not, we contract the step and concentrate sampling around that prefix. This adaptive behavior focuses the LLM's effort on regions of the search space that are more likely to yield high-reward completions.

Figure 2: **Multiple iterations of Alg. 1.** DISC dynamically refines its step sizes across iterations, advancing and contracting the prefix at which it samples from.

Fig.3 illustrates a single iteration of this decision process, where a candidate prefix is either accepted and extended or rejected and contracted. Over multiple iterations, this process yields a full solution

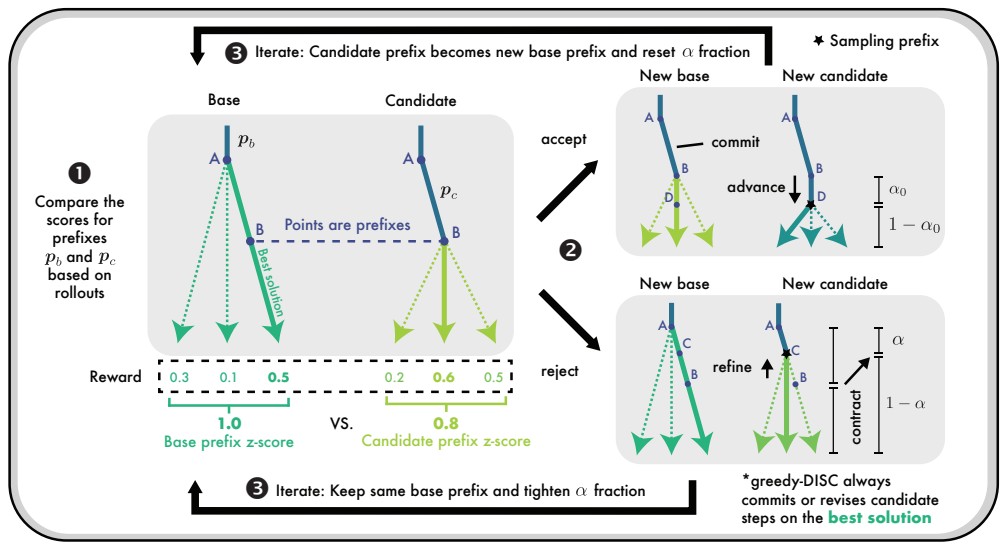

Figure 3: **DISC with Greedy Search.** One iteration of Alg. 1. We start with a base prefix A and a candidate prefix B. We compare the sample statistics of each by evaluating a scoring function (e.g., z-score). If the candidate prefix B demonstrates a higher likelihood of reward improvement compared to continuing from base prefix A, we accept B, commit it as the new base, and extend the candidate to a further step (e.g., BD) on the best sampled solution. If B is not better, we reject it and propose a shorter candidate (e.g., AC), contracting the step size. This process repeats until a new candidate is accepted or all options are exhausted. The algorithm thus adaptively advances or contracts the step size and search horizon based on the relative quality of completions from each prefix.

composed of several such accepted steps. Fig.2 shows how the prefix is incrementally constructed: it displays the number of steps DISC has committed to the prefix, and how the prefix used for sampling dynamically changes over time. Together, these figures highlight the local step-wise decision-making and the global trajectory of prefix refinement throughout the search.

**Assumptions.** DISC makes minimal, broadly applicable assumptions, enabling generality and ease of deployment. It avoids handcrafted prompts, process-level reward models, and domain-specific heuristics. Its only requirement is access to a scalar outcome reward model (ORM) to guide search. In the absence of a ground-truth ORM, self-supervised signals—like LLM critiques or unit tests—serve as effective substitutes [30, 31, 32]. It also assumes the underlying policy $\pi$ can generate continuations from any prefix $p$, a standard feature of decoder-only language models.

## 3.2 DISC with Greedy Search Algorithm

We now describe Alg. 1. The algorithm takes as input an LLM $\pi$, a reward model $R$, prompt $x$, initial partition fraction $\alpha_0$, threshold $\sigma$, and sample budget $N$. It initializes the base prefix as $p_b = x$ and sets $\alpha = \alpha_0$.

At each iteration, the algorithm samples $\pi$ from $p_b$ to generate completions $y^i = p_b \cdot s^i$, computing rewards $R(y^i)$. Sampling stops when the cumulative reward exceeds $\sigma$: $M = \min\{m \in \mathbb{Z}_{>0} \mid \sum_{i=1}^{m} R(y^i) \geq \sigma\}$. This balances sample quantity and quality. The solutions and their rewards are stored in $Y$, and the best suffix $s^*$ and z-score are computed.

A candidate prefix $p_c$ is then formed by appending the first $\alpha$ fraction (token-wise) of $s^*$ to $p_b$. It is accepted if its reward z-score $z_c$ is lower than the current $z_b$. Assuming rewards follow a location-scale family distribution (e.g., Gaussian), a lower z-score implies a higher probability of improvement, since $\Pr(R > \max_{s \in S_{b,c}}(R(s)) = 1 -$

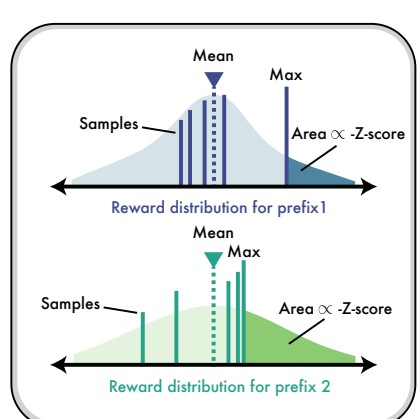

Figure 4: Reward distribution of rollouts sampled from two different prefixes. The probability of sampling a higher rollout from prefix 2 is higher than that of prefix 1.

$\text{CDF}(z_{b,c})$ where CDF is the cumulative distribution function. Fig. 4 displays how we can estimate the probability of improvement from a reward distribution. A location-scale distribution of rewards is supported empirically (see App. C.5).

If the candidate is accepted, the algorithm updates the base prefix $\boldsymbol{p}_b \leftarrow \boldsymbol{p}_c$, resets $\alpha = \alpha_0$, and updates the base z-score and best suffix. If rejected, the partition fraction contracts: $\alpha \leftarrow \alpha \cdot \alpha_0$. This contraction implements DISC's recursive refinement mechanism, focusing the search on more promising regions. The process repeats until the sample budget is exhausted or a correct solution is found.

### 3.3 Analysis of DISC with Greedy Search

DISC exhibits two important properties: (i) the z-score decreases monotonically over the course of algorithm iterations, and (ii) the best candidate solution always has a higher reward than the best base solution. We leverage these properties, together with assumptions about the quality of $\pi$, to establish the following result. We also develop a motivating theoretical example in App. G.2.

**Theorem 1 (Optimality of DISC)** *Consider Alg. 1. Suppose that for some problem $\boldsymbol{x}$, the optimal solution is in the support of $\pi(\cdot \mid \boldsymbol{x})$. Then at some $n > 0$, the base prefix contains EOS token, the algorithm terminates, and this solution is an optimal solution. See App. G for proof.*

**Remark 1** *Our analysis is dependent on a strong policy assumption, which is "reverse engineered" to be the weakest assumption on the policy such that our algorithm terminates at optimality. In other words, this assumption depends on an instance-dependent property that must be checked empirically. The purpose of our analysis is not to provide a universal guarantee, but rather to understand how adaptive decomposition method can control real-time inference compute without sacrificing optimality.*

---

**Algorithm 1** DISC with Greedy Search

**Require:** LLM $\pi$, Reward model $R$, prompt $x$, initial partition fraction $\alpha_0$, negative binomial threshold $\sigma$, sample budget $N$
    // initialize base prefix and current partition fraction
1: $\boldsymbol{p}_b = \boldsymbol{x}$, $\alpha = \alpha_0$, $n = 0$
    // compute base prefix statistics (_b)
2: $S_b = \{\boldsymbol{p}_b \cdot \boldsymbol{s}^i \mid \boldsymbol{s}^i \sim \pi(\cdot|\boldsymbol{p}_b)\}_{i=1}^M$
3: $z_b = \frac{\max_{\boldsymbol{s} \in S_b}(R(\boldsymbol{s})) - \text{mean}_{\boldsymbol{s} \in S_b}(R(\boldsymbol{s}))}{\text{std}_{\boldsymbol{s} \in S_b}(R(\boldsymbol{s}))}$
4: $\boldsymbol{p}_b \cdot \boldsymbol{s}_b^* = \text{argmax}_{\boldsymbol{s} \in S_b} R(\boldsymbol{s})$
5: **while** $n < N$ **do**
      // get candidate prefix (_c)
6:    $\boldsymbol{p}_c = \boldsymbol{p}_b \cdot \text{split}(\boldsymbol{s}_b^*, \alpha)$
      // sample and compute candidate prefix statistics (_c)
7:    $S_c = \{\boldsymbol{p}_c \cdot \boldsymbol{s}^i \mid \boldsymbol{s}^i \sim \pi(\cdot|\boldsymbol{p}_c)\}_{i=1}^M$
8:    $z_c = \frac{\max_{\boldsymbol{s} \in S_c}(R(\boldsymbol{s})) - \text{mean}_{\boldsymbol{s} \in S_c}(R(\boldsymbol{s}))}{\text{std}_{\boldsymbol{s} \in S_c}(R(\boldsymbol{s}))}$
9:    $\boldsymbol{p}_c \cdot \boldsymbol{s}_c^* = \text{argmax}_{\boldsymbol{s} \in S_c} R(\boldsymbol{s})$
10:   $n \leftarrow n + M$
      // accept or reject the candidate prefix
11:   **if** $z_c < z_b$ or $|\boldsymbol{s}_b^*| \leq 1$ **then**
12:      $\boldsymbol{p}_b \leftarrow \boldsymbol{p}_c$, $\boldsymbol{s}_b^* \leftarrow \boldsymbol{s}_c^*$,
13:      $\alpha \leftarrow \alpha_0$, $z_b \leftarrow z_c$
14:   **else**
15:      $\alpha \leftarrow \alpha_0 \alpha$
16:   **end if**
17: **end while**
18: **yield** $\boldsymbol{p}_b \cdot \boldsymbol{s}_b^*$

---

### 3.4 DISC for Plug-and-Play with Search Algorithms

In the previous section, we presented DISC with greedy search as a complete algorithm. However, the core of DISC is the decomposition that controls from which token prefixes to query the model and dynamically adjusts step sizes, which can be plug-and-played with other search algorithms like Beam search and MCTS. The DISC EXPANDNODE and MAKECHILDREN are presented in Alg. 3. These methods are conceptually similar to the DISC with greedy search, except that in the general search context, each node stores its own base prefix, and so when a node is expanded it computes its candidate prefix from the suffix to that node instead of from the best existing suffix. During the expansion operator, multiple sets of children are generated, but only the children from the final prefix are kept for search.

### 3.5 Example Decomposition

Using a sampling budget of $N = 100$ LLM calls, the decomposition of a representative MATH problem is shown below. Each step is enclosed in brackets and color-coded based on the z-score of

sampling a better suffix conditioned on the current prefix. Brighter colors indicate lower z-scores, signaling higher importance.

---

**DISC example decomposition**

[ Let the length of the rectangle be $l$ ] [ and the width of the rectangle be $w$. Since the perimeter of the rectangle is 24 inches, we have that $2l + 2w = 24$, so $l + w = 12$. We wish to maximize the area of the rectangle, ] [ which ] [ is $A = lw$. Let $l = 12 - w$ and plug into the area:

$$A = (12 - w)w \Rightarrow \qquad A = 12w - w^2$$

Now, we differentiate $A$ with respect to $w$:

$$A'(w) = 12 - 2w$$

We wish to maximize $A$, so we set $A'(w) = 0$, and solve for $w$:

$$12 - 2w = 0 \Rightarrow \qquad w = 6$$

Since $l = 12 - w$, we have that $l = 12 - 6 = 6$. Therefore, the area of the rectangle is $A = lw = 6 \cdot 6 = \boxed{36}$. ]

---

DISC identifies the **first step** as highly important, which aligns with intuition—early reasoning forms the foundation for all subsequent steps. In contrast, the **final step**, although large, is marked as low-importance, indicating that DISC allocated minimal compute toward refining it. This suggests that once earlier reasoning is fixed, there is limited opportunity for improvement in the final conclusion using additional sampling. Interestingly, the **third step**, beginning with *"which"*, is assigned high importance and receives substantial attention from DISC. This step appears to act as a pivotal decision point that shapes the direction of the remaining solution. This observation supports the idea that certain tokens or sub-sequences function as critical reasoning forks or pivots—consistent with findings from prior work [33, 34].

---

**Autoregressive models require autoregressive decomposition**

While transition words such as 'which', 'therefore', 'wait', etc. may not appear significant to human readers, our decomposition frequently identifies them as critical decision points for autoregressive LLMs trained on next-token prediction, where selecting a different token at these junctures can substantially alter the downstream reasoning and final outcome. Therefore, it is essential for inference algorithms to allocate more compute toward sampling at these steps, and to identify these steps automatically through LLM data rather than through human design.

---

## 4 Experimental Results

### 4.1 Main Results

**Benchmarks.** We evaluate DISC on three benchmarks: **APPS**, **MATH**, and **LiveCodeBench**, to assess its impact on inference scaling for both coding and reasoning. **APPS** [18] consists of 5000 competitive programming problems across three difficulty levels, with the competition-level subset being the hardest. We evaluate on a 200-problem subset due to computational constraints. **MATH** [35] comprises 12,500 math problems. Since the ground-truth verifier provides only binary rewards, we use a pretrained ORM [36], trained via the method in [37], with Llama-3.1-8B-Instruct as the base model. We test on a 500-problem subset (**MATH500**), identical to prior work [37, 23]. **LiveCodeBench** [38] is a continuously updated dataset from Leetcode, AtCoder, and CodeForces, ensuring LLMs have not been exposed to test problems. We evaluate on the 108 problems uploaded between 10/01/2024 and 12/01/2024 to prevent contamination.

**Baselines.** We compare DISC against three prior decomposition methods: **TokenSplit** (token-level decomposition), **LineSplit** (newline-based decomposition), and **BoN** (treating the entire solution as a single step). These are all standard and the most commonly used methods (see Sec. 2.2).

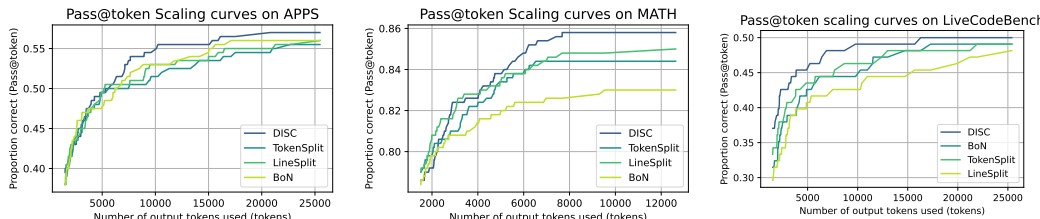

Figure 5: **Token-level comparisons across benchmarks** using gpt-4o-mini. (Left) APPS competition level (Middle) MATH500 (Right) LiveCodeBench. DISC achieves superior inference scaling over baselines on all three benchmarks.

**Metrics.** We evaluate two key metrics: **Pass@k**, the proportion of problems solved within a sample budget $k$, and **Pass@token**, the proportion solved within a given token budget. Note that $k$ refers to the sample budget, *not thousands of samples*, and error proportion refers to proportion not solved. We use $\alpha_0 = 0.15, \sigma = 1.0$, and temperature $\tau = 0.2$ by default for DISC unless otherwise specified.

**Performance.** Across all benchmarks, DISC consistently delivers stronger performance and better scaling under both fixed token budgets (Fig. 5) and fixed sample budgets (Fig. 8). For example, on APPS, the pass@10 error proportion decreases from 0.50 to 0.475; on MATH500, from 0.15 to 0.14; and on LiveCodeBench, from 0.57 to 0.51. These correspond to a **5.0%, 6.7%, and 10.5% reduction in error** relative to the best baseline, respectively. These improvements are particularly meaningful on more challenging benchmarks, where performance gains are harder to achieve, demonstrating DISC 's effectiveness in guiding search toward high-reward regions. Extended results and analyses are provided in App. D.1, D.4, and D.5.

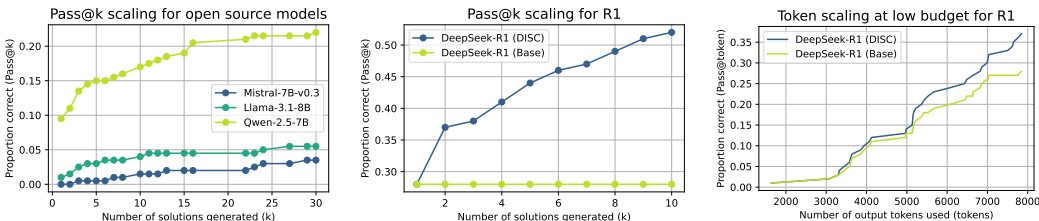

Figure 6: **Inference scaling for different models, including reasoning models, on APPS. (Left)** Pass@k for open-source models. Colors correspond to different models, with Pass@1 being the base model performance. **(Middle)** Pass@k for R1, with green line as base Pass@1 performance. **(Right)** Pass@token for R1. DISC almost **doubles** base performance across all models.

**Model Generality.** DISC also provides substantial gains across a range of models, including open-source LLMs. For instance, it improves LLaMA's pass@10 rate from 0.01 to 0.04—a **300% relative increase**. Similarly, it boosts Mistral's performance from 0.0 to 0.02, and Qwen's from 0.095 to 0.17, representing a **79% relative increase**. As shown in Fig. 6 (left), these improvements hold consistently across sampling budgets, including in low-budget regimes. This demonstrates DISC 's general applicability and effectiveness even for weaker or resource-constrained models. Additional results and analyses are provided in App. C.3.

**Reasoning Models.** DISC also yields substantial gains when applied to strong long-form chain-of-thought (CoT) reasoning models such as R1 [15, 39]. As shown in Fig. 6 (middle), DISC improves accuracy by over **85%** relative to the base R1 model using just 10 samples. Notably, Fig. 6 (right) shows that even under a constrained token budget—matched to that of a single sample from the base model—DISC still achieves over a **33% relative improvement**. This demonstrates that DISC not only scales well with more samples but is also highly effective at identifying and prioritizing critical reasoning steps, leading to stronger performance even in low-resource settings.

**Computational Overhead.** DISC introduces negligible runtime overhead compared to standard decoding baselines, as shown in Fig. 7. The vast majority of compute time—over 90% across all settings—is still dominated by LLM token sampling, with only a minor fraction spent on auxiliary

operations such as candidate management, z-score normalization, and recursive branching. Despite its dynamic control flow, DISC maintains a runtime composition nearly identical to that of methods like BoN and LineSplit. Moreover, because DISC achieves higher success rates with fewer tokens (Fig. 5), its effective runtime per correct solution is even lower. In practice, this makes DISC both algorithmically efficient and computationally scalable, offering improved search performance without additional inference cost. We provide a detailed runtime discussion in App. D.6.

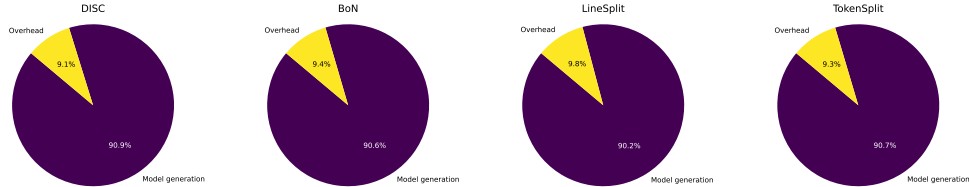

Figure 7: **Percentage of runtime spent on overhead vs LLM token generation.** DISC does not increase the runtime overhead significantly.

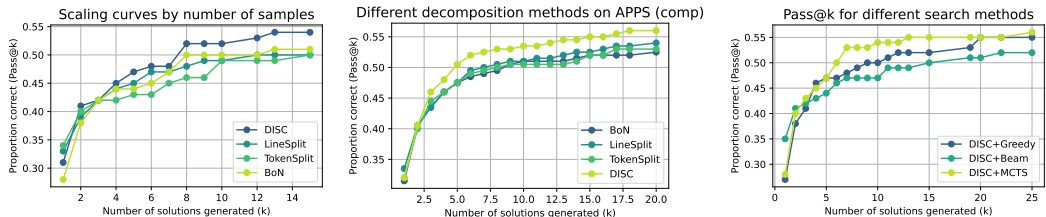

Figure 8: **Comparison of Pass@k performance on APPS using gpt-4o-mini.** (Left) self generated validation tests, (Middle) with ground truth tests, (Right) with different search methods.

**Self-Generated Validation.** We further evaluate DISC in a self-generated validation setting, where ground-truth reward models or curated tests are unavailable [30, 40, 41]. Here, the LLM generates its own unit tests from the problem description, which serve as a **proxy reward model** for evaluating candidate solutions. This setup offers a *scalable alternative* to costly manual test curation in real-world code generation tasks.

As shown in Fig. 8, DISC scales effectively under this protocol, achieving a **54% relative improvement** over the base model. This demonstrates that DISC leverages decomposition-based reasoning to produce higher-quality code even when supervision is noisy or incomplete.

However, self-generated validation has limitations: generated tests may be incomplete, inconsistent, or narrow in coverage, potentially biasing performance estimates. While these issues do not affect the observed scaling trends, improving test reliability and robustness remains an important direction. Additional details and results are provided in App. D.3.

**Search.** We demonstrate that search strategies such as **MCTS** and **beam search** can be naturally integrated with DISC using the approach described in Sec. 3.4. As shown in Fig. 46, greedy search tends to explore deeper partitions within the same search budget due to its myopic nature, while MCTS and beam search explore more diverse but shallower paths. Despite similar depth, **MCTS** outperforms beam search by allocating its search budget more strategically—focusing exploration on more promising candidates—resulting in superior overall performance, as seen in Fig. 8. Furthermore, unlike greedy search, which irrevocably commits to prefixes as they are accepted, MCTS maintains flexibility by exploring committing multiple candidate prefixes in parallel. Additional details and analysis are provided in App. F.

### 4.2 Ablation Studies

**Temperature.** Typically, inference scaling methods achieve optimal performance at temperatures around 0.6–0.8, as increased temperature promotes sample diversity [42]. Surprisingly, however, DISC performs *better at lower temperatures*, as shown in Fig. 9. This trend is in stark contrast to

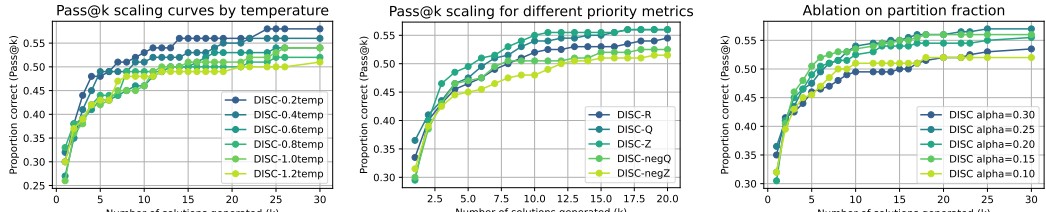

Figure 9: **Ablation study of DISC on APPS with gpt-4o-mini.** (Left) Effect of temperature: unlike BoN and other inference scaling methods, DISC achieves higher performance at lower temperatures. (Middle) effect of acceptance method (Right). Effect of partition fraction $\alpha_0$: The range $0.15 \leq \alpha_0 \leq 0.25$ appears optimal.

BoN (Fig. 16), where higher temperatures are generally beneficial. We believe this phenomenon arises because DISC depends on estimating the z-score at each step using sample statistics. Lower temperatures reduce sample variance, leading to more reliable estimates, which in turn improves step selection. This is further supported by Fig. 14, which shows that lower temperatures yield lower standard deviations per step, indicating increased sampling consistency. Additional details and analyses can be found in App. C.1.

**Acceptance Method.** We perform an ablation study to evaluate whether using the z-score is an effective criterion for accepting candidate prefixes. Specifically, we compare our standard z-score-based acceptance method, DISC-Z, against four alternative baselines: DISC-R, which accepts candidates uniformly at random; DISC-Q, which accepts if the candidate prefix has a lower mean value; DISC-negQ, which accepts if the candidate has a higher mean; and DISC-negZ, which accepts if the candidate has a higher z-score (rather than a lower one). As shown in Fig. 9, the choice of acceptance criterion substantially influences performance. Among all methods, DISC-Z achieves the highest performance, while DISC-negZ performs worse than random selection, underscoring the importance of prioritizing candidates with a higher probability of improvement. Additional details and analysis are in App. C.2.

**Partition Fraction $\alpha_0$.** As shown in Fig. 9 and Fig. 25, performance is highest when the partition fraction lies in the range $0.15 \leq \alpha_0 \leq 0.25$. Smaller values of $\alpha_0$ generally yield better results because they lead to more conservative proposals—i.e., shorter candidate prefixes. This conservatism is beneficial due to the high cost of prematurely committing to a suboptimal prefix: once a candidate prefix is accepted (in greedy-DISC), it becomes fixed and cannot be revised. By keeping candidate prefixes short, the algorithm retains more flexibility to correct course in future steps. Additional analysis is provided in App. C.4.

### 4.3 Analysis and Interpretation

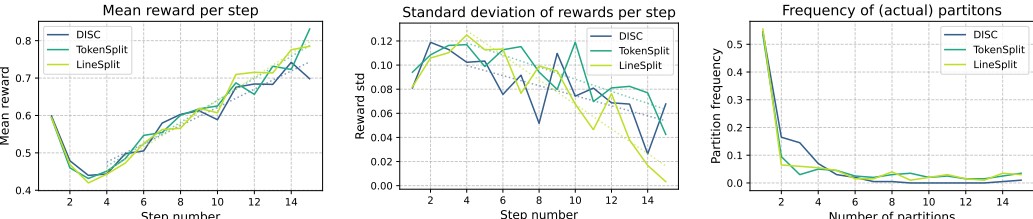

Figure 10: **Analysis of decomposition methods.** Dotted lines fit a linear model to indicate the trend. **(Left)** Average reward per step: From step 3 onward, higher step counts strongly correlate with increased average reward, demonstrating the effectiveness of decomposition. The dip between steps 1 and 3 likely occurs because simple problems are solved early, preventing further search. **(Middle)** Standard deviation of rewards per step: Decomposition reduces sampling variance, improving precision at deeper search depths. **(Right)** Frequency of the number of partition steps that the search algorithm committed to the prefix during the search process.

Our results strongly suggest that decomposition—whether line-based, token-based, or DISC —consistently improves sample quality. Fig. 10 (left and middle) shows how the mean and variance of sampled rewards evolve with the **step number**, i.e., the number of steps committed to the prefix during search. As the step number increases, the mean reward improves, indicating that longer committed prefixes lead to higher-quality solutions. At the same time, the variance of rewards

decreases, suggesting that committing to a longer prefix also improves the precision of sampling. These trends highlight the benefits of finer-grained decomposition and incremental commitment in guiding the search process more effectively.

Furthermore, DISC achieves higher performance using fewer committed prefix steps—and thus fewer sampling stages—under a fixed sampling budget. Fig. 10 (right) shows the distribution of **actual partitions**, i.e., the number of steps effectively committed under a finite budget. As shown, DISC typically requires only 1–5 actual partition steps, whereas other methods commit to significantly more. This indicates that DISC is more efficient at identifying high-impact prefixes, enabling better performance with fewer sampling decisions.

**Limitations.** While DISC demonstrates strong empirical performance, several limitations remain (see Appendix E for a detailed discussion). In particular, DISC assumes access to a reasonably accurate reward model. For example, in code generation tasks, this requires that either ground-truth validation tests are available or the model can self-generate sufficiently reliable ones. Addressing these limitations—through improved test generation, learned verifiers, and more robust evaluation protocols—presents promising directions for future work.

> **Decomposition and Sample Quality**
>
> DISC enables more efficient exploration by identifying high-impact prefixes with fewer steps. Incremental prefix commitment not only improves sample quality—yielding higher average rewards—but also reduces reward variance, leading to more stable and reliable outputs under a fixed sampling budget.

## 5 Related Work

**Inference scaling.** Inference scaling has emerged as a dominant paradigm, driven by the introduction of o1- and r1-like chain-of-thought reasoning models [5, 6, 43, 44]. Several works examine the trade-off between inference compute and training compute [45, 46]. LLM inference often relies on decomposing complex problems into intermediate reasoning steps, as seen in chain-of-thought (CoT) prompting [47, 48, 49] and its variants [50, 51, 52, 53]. We extend inference scaling by introducing a new approach for adaptive compute allocation [43, 54, 55].

**LLM reasoning and code generation.** LLM reasoning and code generation are central tasks for inference scaling. Evolutionary inference scaling methods have been explored in program generation [56, 57, 58, 59, 60]. Domain-specific decomposition strategies have been applied in code generation, such as function-based decomposition [61, 62, 63]. More broadly, decomposition often involves prompting LLMs to generate subtask completions [64, 65, 66], which differs from methods that refine a single LLM generation.

**Reinforcement learning and Monte Carlo methods.** Unlike standard RL, our setting resembles a search problem where the goal is to identify the single highest-reward path. Nested Monte Carlo search can accelerate optimal pathfinding [67]. Under the bandit setting, this can be formulated as identifying the arm with the highest *maximum* reward rather than the highest mean reward [68, 69].

## 6 Conclusion

We introduce DISC, a dynamic decomposition framework that adaptively partitions solution steps based on first order statistics that capture potential for improvement, improving inference scaling by directing compute toward critical steps while balancing exploration and resource allocation. DISC naturally integrates with search-based methods such as MCTS and beam search, further enhancing performance. It also identifies challenging steps for LLMs, aiding curriculum learning, fine-tuning, and dataset augmentation. By dynamically adjusting partitioning and step sizes based on available compute, DISC enables more adaptive and efficient reasoning in large language models, with broad implications for both training and inference optimization.

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

# Contents

# A  Code implementation of DISC

To aid reproducibility and practical adoption, we provide Python-style pseudocode in this section that closely mirrors our actual implementation. The pseudocode abstracts away low-level engineering details while preserving the core logic, function names, and control flow used in our codebase. This alignment ensures that readers can easily translate the pseudocode into a working implementation or modify it for their own use cases. All key components of the DISC algorithm—including dynamic step refinement, z-score-based acceptance, and integration with search strategies—are reflected in this pseudocode with minimal deviation from the actual code.

```python
def dynamic_decomposition(problem, model, reward_model, split_str, complete_solution, fraction,
    solution_budget, split_metric, stop_threshold=-float("inf"), stop_sum_score=1.0,
    stop_if_solved=False, ):
    """
    Decomposes the solution using a dynamic binary search approach

    Args:
        problem (Problem): The problem to solve
        model (Model): The model to use for generation
        reward_model (function): The reward model to use for scoring
        split_str (function): The function to use for splitting a string
        complete_solution (function): The function to use for completing a solution
        fraction (float): The fraction to split the string
        solution_budget (int): The maximum number of solutions to generate
        split_metric (function): The metric to use for splitting
        stop_threshold (float): The threshold to stop splitting
        stop_sum_score (float): The sum score to stop generating completions
        stop_if_solved (bool): Whether to stop if the problem is solved
    """

    # Initialize results and decomposition steps
    decomp_return = {
        "generated_solutions": [],
        "decomposition": []
    }

    while len(decomp_return["generated_solutions"]) < solution_budget:
        # Combine all previous steps into an intermediate solution
        intermediate_solution = "".join([step["step_str"] for step in decomp_return["decomposition"
            "]])
        new_scores = []
        best_solution = None
        best_completion = None
        best_score = -float("inf")
        sum_score = 0.0

        # 1) Generate completions until we generate enough samples to estimate the split metric
        while sum_score < stop_sum_score:
            proposed_completion = complete_solution(problem, intermediate_solution, model)
            proposed_solution = intermediate_solution + proposed_completion
            decomp_return["generated_solutions"].append(proposed_solution)

            # Update scores
            proposed_score = reward_model(proposed_solution)
            new_scores.append(proposed_score)
            sum_score += proposed_score

            # Track the best solution
            if proposed_score > best_score:
                best_solution = proposed_solution
                best_score = proposed_score
                best_completion = proposed_completion

            # Stop early if problem is solved
            if stop_if_solved and proposed_score >= 1.0:
                decomp_return["decomposition"].append({"step_str": proposed_completion})
                return decomp_return

        new_metric = split_metric(new_scores)
        last_metric = decomp_return["decomposition"][-1]["metric"] if decomp_return["decomposition
            "] else None

        # Determine the split target. We always split the step with the highest metric
        is_split_new_step = last_metric is None or new_metric >= last_metric
        split_target = decomp_return["decomposition"][-1]["step_str"] if not is_split_new_step
            else best_completion
```

```python
        # 3) Attempt to split the target
        split_result = split_str(split_target, fraction)
        if not split_result: # If we can't split the target, we're done
            decomp_return["decomposition"].append({"step_str": best_completion, "metric":
                new_metric})
            return decomp_return

        # Update decomposition based on split
        part1, part2 = split_result
        if is_split_new_step:
            decomp_return["decomposition"].append({"step_str": part1, "metric": new_metric})
            # Stopping condition based on threshold
            if new_metric < stop_threshold:
                decomp_return["decomposition"].append({"step_str": part2})
                return decomp_return
        else:
            decomp_return["decomposition"][-1] = {"step_str": part1, "metric": last_metric}

    return decomp_return
```

## B   Pseudocode for DISC

**Algorithm 2** Dynamic Decomposition

1: **Input:** Problem instance $x$, reward model $r$, partition function $f$, LLM policy model $\pi$, partition fraction $\alpha$, solution budget $B$, priority metric $h$, metric stopping precision $\theta$, sampling stopping threshold $\sigma$, is inference mode $\mathbf{b}_{\text{inference}}$
2: **Output:** Final decomposition $D$
3: Initialize $D \leftarrow \{\text{generated\_solutions} : \varnothing, \text{decomposition} : \varnothing\}$
4: # Decompose the solution recursively until we reach the desired precision $\theta$ or run out of budget $B$
5: **while** $|D.\text{generated\_solutions}| < B$ **do**
6:     $y_{\text{intermediate}} \leftarrow \text{Concatenate}([\text{step.step\_str} \ \forall \ \text{step} \in D.\text{decomposition}])$
7:                                          $\triangleright$ Concatenate previous steps to form intermediate solution
8:     $R_{\text{new}} \leftarrow \varnothing$                                         $\triangleright$ Record rewards of completions
9:     $\text{best}.y_{\text{final}} \leftarrow \text{None}, \text{best}.y_{\text{completion}} \leftarrow \text{None}, \text{best}.r \leftarrow -\infty$    $\triangleright$ Track the best completion
10:     # Step 1: Generate completions until we have enough samples to estimate the splitting metric. Here we use a geometric sampling distribution
11:     **while** $\text{sum}(R_{\text{new}}) < \sigma$ **do**
12:         $y_{\text{completion}} \leftarrow \pi(\cdot|x, y_{\text{intermediate}})$
13:         $y_{\text{proposed}} \leftarrow y_{\text{intermediate}} \oplus y_{\text{completion}}$
14:         Append $y_{\text{proposed}}$ to $D.\text{generated\_solutions}$
15:         $r_{\text{proposed}} \leftarrow r(y_{\text{proposed}})$
16:         Append $r_{\text{proposed}}$ to $R_{\text{new}}$
17:         **if** $r_{\text{proposed}} > \text{best}.r$ **then**
18:             $\text{best}.y_{\text{final}} \leftarrow y_{\text{proposed}}, \text{best}.y_{\text{completion}} \leftarrow y_{\text{completion}}$
19:             $\text{best}.r \leftarrow r_{\text{proposed}}$
20:         **end if**
21:         **if** $\mathbf{b}_{\text{inference}}$ **and** $r_{\text{proposed}} = 1.0$ **then**
22:             Append $\{\text{step\_str} : y_{\text{completion}}\}$ to $D.\text{decomposition}$
23:             **Return** $D$                      $\triangleright$ Exit if problem is solved
24:         **end if**
25:     **end while**
26:     # Step 2: Compute splitting metric
27:     $z_{\text{new}} \leftarrow h(R_{\text{new}})$
28:     $z_{\text{last}} \leftarrow D.\text{decomposition}[-1].z$ **if** $D.\text{decomposition} \neq \varnothing$ **else** $-\infty$
29:     # Step 3: Split the step with the higher metric
30:     $\mathbf{b}_{\text{split new step}} \leftarrow z_{\text{new}} \geq z_{\text{last}}$
31:     $y_{\text{target step}} \leftarrow \text{best}.y_{\text{completion}}$ **if** $\mathbf{b}_{\text{split new step}}$ **else** $D.\text{decomposition}[-1].\text{step\_str}$
32:     $y_1, y_2 \leftarrow f(y_{\text{target step}}, \alpha)$
33:     **if** $y_1 = \text{None}$ **or** $y_2 = \text{None}$ **then**
34:         Append $\{\text{step\_str} : y_{\text{completion}}, \text{metric} : z_{\text{new}}\}$ to $D.\text{decomposition}$
35:         **Return** $D$                  $\triangleright$ Exit if we cannot do a finer split
36:     **end if**
37:     **if** $\mathbf{b}_{\text{split new step}}$ **then**
38:         Append $\{\text{step\_str} : y_1, \text{metric} : z_{\text{new}}\}$ to $D.\text{decomposition}$       $\triangleright$ Add new step
39:         **if** $z_{\text{new}} < \theta$ **then**
40:             Append $\{\text{step\_str} : y_2\}$ to $D.\text{decomposition}$
41:             **Return** $D$            $\triangleright$ Exit if all metrics are smaller than precision
42:         **end if**
43:     **else**
44:         $D.\text{decomposition}[-1] \leftarrow \{\text{step\_str} : y_1, \text{metric} : z_{\text{last}}\}$        $\triangleright$ Split last step
45:     **end if**
46: **end while**
47: **Return** $D$

# C   Ablation studies

## C.1   Ablation on Temperature

We conduct an ablation study to analyze the effects of temperature on DISC and BoN. Temperature controls the randomness of token sampling in autoregressive models, influencing both exploration and consistency. Higher temperatures encourage more diverse outputs, whereas lower temperatures yield more deterministic generations. To examine its impact, we evaluate DISC and BoN on a 100-problem subset of APPS (the first 100 problems) using gpt-4o-mini.

Figure 11 presents the Pass@token scaling curve for DISC across different temperatures. The results indicate that lower temperatures lead to improved performance, as DISC benefits from more deterministic step selection. Unlike BoN, which relies on broad solution sampling, DISC dynamically refines steps, making stable token probabilities advantageous.

Figure 12 illustrates the frequency of actual partitions made by DISC at different temperatures. As temperature increases, the number of partitions fluctuates more, suggesting that high temperature introduces instability in step selection. Lower temperatures provide more structured decomposition, reducing unnecessary subdivisions.

In Figure 13, we visualize the mean reward per step. The trend shows a linear increase in reward as step number grows, demonstrating that deeper decomposition results in progressively better solutions. This reinforces that DISC effectively allocates computation towards refining difficult steps.

The mean standard deviation per step is shown in Figure 14. Lower temperatures yield lower standard deviations, confirming that DISC benefits from reduced variability in sample quality. This consistency allows for more reliable prioritization of difficult steps, enhancing overall inference efficiency.

For comparison, Figure 16 and Figure 15 display Pass@token and Pass@k scaling curves for BoN across different temperatures. Unlike DISC, BoN achieves peak performance at a temperature around 0.6-0.8, balancing diversity and consistency. Higher temperatures increase exploration but degrade precision, while lower temperatures hinder sample diversity, reducing the probability of obtaining high-quality completions.

These findings highlight the fundamental difference between DISC and BoN: DISC benefits from lower variance and stable decomposition, while BoN relies on broader exploration facilitated by moderate temperature settings. As a result, optimal temperature settings differ significantly between these methods, with DISC favoring deterministic sampling and BoN requiring a balance between diversity and coherence.

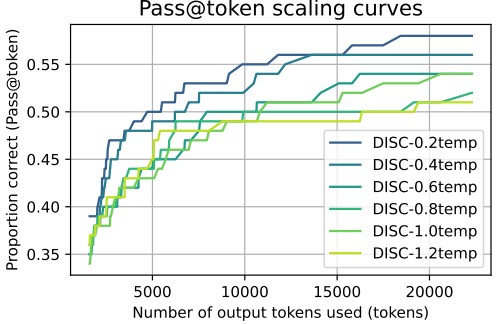

Figure 11: **Pass@token scaling curve for different temperatures on APPS using gpt-4o-mini**. The lower the temperature, the stronger the DISC performance.

Figure 12: **Partition frequency of DISC with different temperatures on APPS using gpt-4o-mini**.

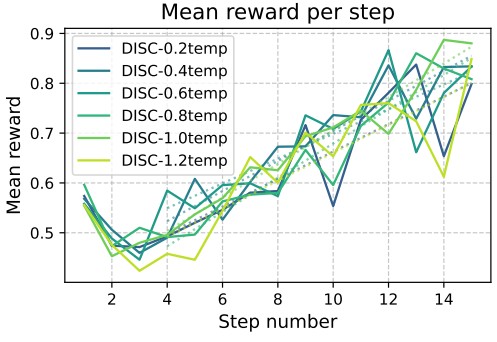

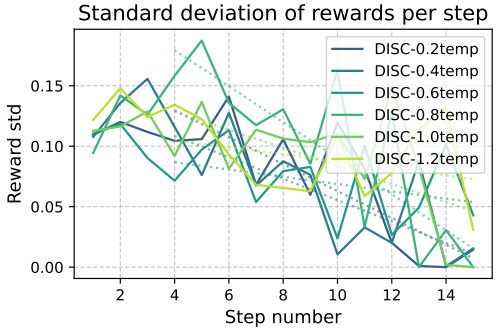

Figure 13: **Mean reward per step of DISC with different temperatures on APPS using gpt-4o-mini**. The mean reward scales linearly with step number.

Figure 14: **Mean standard deviation per step of DISC with different temperatures on APPS using gpt-4o-mini**. Lower temperature means lower average standard deviation.

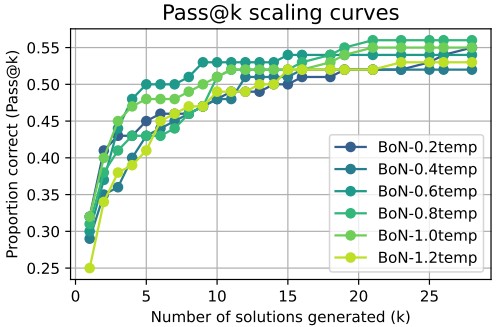

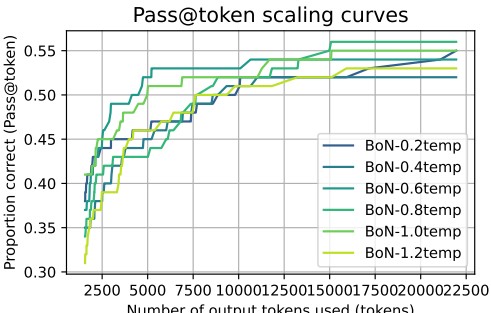

Figure 15: **Pass@k scaling curve for different temperatures on APPS using gpt-4o-mini for BoN**. A temperature around 0.6–0.8 leads to the best performance and balance between diversity and consistency.

Figure 16: **Pass@token scaling curve for different temperatures on APPS using gpt-4o-mini for BoN**. A temperature around 0.6–0.8 leads to the best performance and balance between diversity and consistency.

## C.2 Acceptance Method

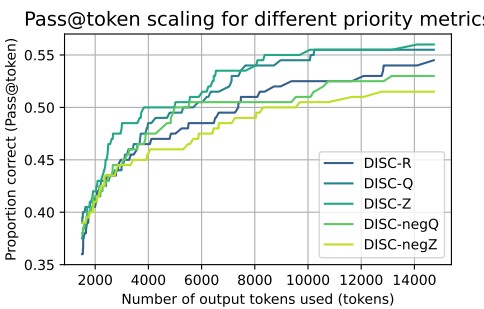

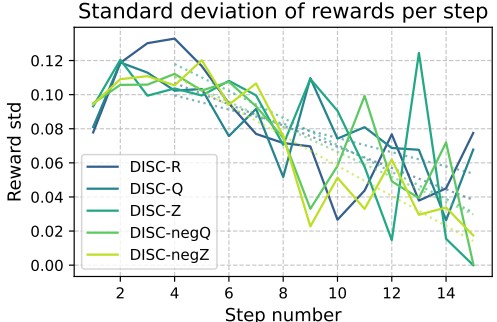

Figure 17: **Token level comparison of different priority metrics on DISC in the APPS setting with gpt-4o-mini.** Both Q and Z based priority metrics perform well.

Figure 18: **Partition frequency of DISC with different priority metrics on APPS using gpt-4o-mini.**

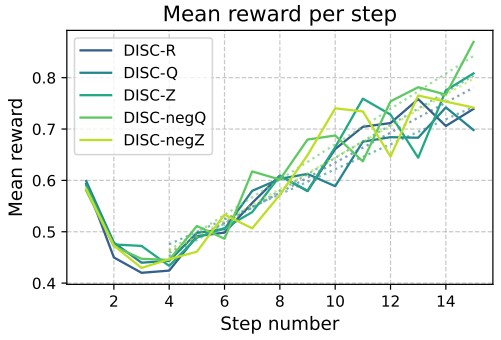

Figure 19: **Mean reward per step of DISC with different priority metrics on APPS using gpt-4o-mini**. All metrics display strong correlation between step depth and the mean reward.

Figure 20: **Mean standard deviation per step of DISC with different priority metrics on APPS using gpt-4o-mini**. All metrics display correlation between step depth and the standard deviation.

We analyze the impact of different **acceptance methods** on DISC's performance. These methods govern whether a candidate prefix is accepted for further decomposition, directly influencing efficiency, stability, and final solution quality.

We begin by introducing notation to characterize the distribution of rewards obtained when sampling completions from a given prefix $\boldsymbol{p}$. Let $R_{\boldsymbol{p}}$ denote the random variable of the reward of a completion sampled from the LLM policy $\pi$ conditioned on prefix $\boldsymbol{p}$:

$$R_{\boldsymbol{p}} := R(\boldsymbol{p} \cdot \boldsymbol{s}), \quad \text{where } \boldsymbol{s} \sim \pi(\cdot \mid \boldsymbol{p}).$$

Let $F_{\boldsymbol{p}}$, $\mu_{\boldsymbol{p}}$, and $\sigma_{\boldsymbol{p}}$ be the cumulative distribution function (CDF), mean, and standard deviation of $R_{\boldsymbol{p}}$, respectively.

To estimate these quantities in practice, we sample $M$ completions from the policy:

$$Y_{\boldsymbol{p}} = \left\{ \left( \boldsymbol{p} \cdot \boldsymbol{s}^i, \, R(\boldsymbol{s}^i) \right) \mid \boldsymbol{s}^i \sim \pi(\cdot \mid \boldsymbol{p}) \right\}_{i=1}^{M}.$$

From these samples, we compute the empirical mean and standard deviation:

$$\mu_{\boldsymbol{p}} = \frac{1}{M} \sum_{i=1}^{M} R(\boldsymbol{s}^i), \quad \sigma_{\boldsymbol{p}} = \sqrt{\frac{1}{M} \sum_{i=1}^{M} \left( R(\boldsymbol{s}^i) - \mu_{\boldsymbol{p}} \right)^2}.$$

Let $r_{\boldsymbol{p}}^{(1)} = \max_i R(\boldsymbol{s}^i)$ denote the sample maximum reward observed for prefix $\boldsymbol{p}$.

We evaluate DISC on the first 200 competition-level APPS problems using `gpt-4o-mini` at a fixed temperature of **0.8**. The following acceptance methods are compared:

- **DISC-Z**: accepts if the candidate prefix $\boldsymbol{c}$ has a lower z-score than the base prefix $\boldsymbol{b}$.

- **DISC-Q**: accepts if the candidate prefix has a lower mean reward, i.e., $\mu_{\boldsymbol{c}} < \mu_{\boldsymbol{b}}$.

- **DISC-negZ**: accepts if the candidate has a higher z-score.

- **DISC-negQ**: accepts if the candidate has a higher mean reward.

- **DISC-R**: accepts a candidate uniformly at random.

**Z-score Based Acceptance (DISC-Z).**   For a base prefix $\boldsymbol{b}$ and candidate prefix $\boldsymbol{c}$, we estimate the z-score of the sample maximum $r_{\boldsymbol{c}}^{(1)}$ as:

$$z_{\boldsymbol{c}} = \frac{r_{\boldsymbol{c}}^{(1)} - \mu_{\boldsymbol{c}}}{\sigma_{\boldsymbol{c}}}, \quad \text{so } \mathbb{P}[R_{\boldsymbol{c}} > r_{\boldsymbol{c}}^{(1)}] = 1 - F_{\boldsymbol{c}}(r_{\boldsymbol{c}}^{(1)}).$$

We accept $\boldsymbol{c}$ if $z_{\boldsymbol{c}} < z_{\boldsymbol{b}}$. A lower z-score implies a greater tail probability mass and thus a higher chance of improvement.

**Q-based Acceptance (DISC-Q).**   This method accepts $\boldsymbol{c}$ if its sample mean is lower than that of $\boldsymbol{b}$:

$$\mu_{\boldsymbol{c}} < \mu_{\boldsymbol{b}}.$$

While simple, this method compares absolute expected values without accounting for reward variance, making it less robust to noise in $\pi$'s samples.

**Empirical Comparison.**   Figure 17 shows token-level performance across methods. DISC-Z significantly outperforms alternatives, highlighting the value of normalizing reward advantage by variance.

Figure 18 shows that DISC-Z produces fewer but more meaningful partitions, suggesting better allocation of compute.

Figure 19 shows that DISC-Z achieves sharper reward gains early on, indicating better prioritization of impactful refinements.

Figure 20 reports step-wise reward variance. DISC-Z yields the most stable and consistent improvements.

> **Why Z-score Works Best**
>
> Z-score based acceptance balances the mean and variance of reward samples, estimating the probability of improvement in a statistically grounded way. This leads to more reliable and efficient decomposition decisions.

Overall, our results confirm that **acceptance method design is critical for dynamic decomposition**. Among all tested methods, DISC-Z consistently delivers the best performance and compute efficiency.

## C.3 Model Ablation

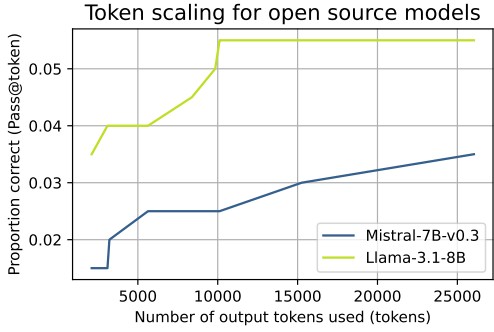

Figure 21: **Pass@token scaling curve for open source models with DISC on APPS**. DISC also demonstrates strong performance gains with open source models.

Figure 22: **Partition frequency of DISC with open source models on APPS**.

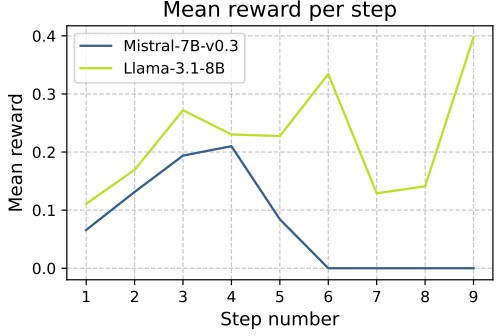

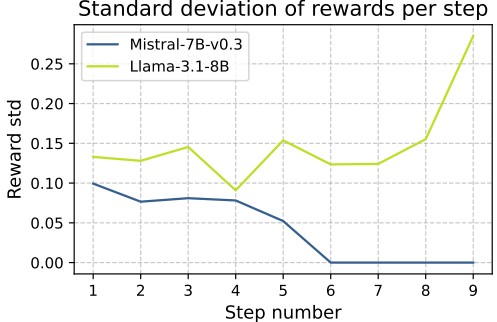

Figure 23: **Mean reward per step of DISC with open source models on APPS**.

Figure 24: **Mean standard deviation per step of DISC with open source models on APPS**.

We investigate how different LLMs perform when used with DISC on 200 competition-level APPS problems, given a sample budget of 30. The groundtruth reward model was used to evaluate correctness, and all models were set to a temperature of 0.8. Due to the challenging nature of the benchmark, open-source models struggled to achieve strong performance independently. However, when paired with DISC, their performance significantly improved.

Figure 21 presents the Pass@token scaling curve for open-source models using DISC. The results demonstrate that DISC substantially enhances the capabilities of these models, closing the gap between them and proprietary alternatives.

Figure 22 visualizes the partition frequency of DISC with different open-source models. Compared to their standalone performance, the use of DISC led to more structured and effective decomposition, highlighting its adaptability to different architectures.

The mean reward per step is shown in Figure 23. Similar to prior findings, we observe that deeper decomposition leads to increasingly higher rewards. Notably, even lower-capacity models benefit from DISC 's ability to iteratively refine their solutions.

Finally, Figure 24 presents the mean standard deviation per step. With DISC, the variance in performance is significantly reduced, resulting in more stable and reliable inference.

Overall, these findings emphasize that DISC is a robust framework capable of enhancing inference performance across diverse LLMs, particularly those with limited standalone capabilities.

## C.4 Ablation on Partition Fraction $\alpha$

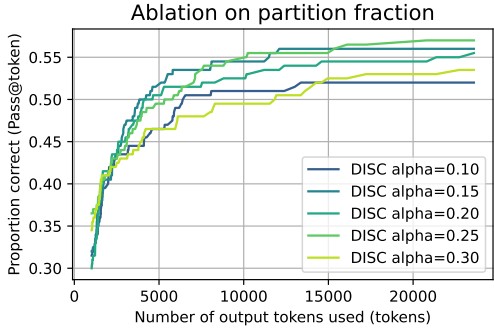 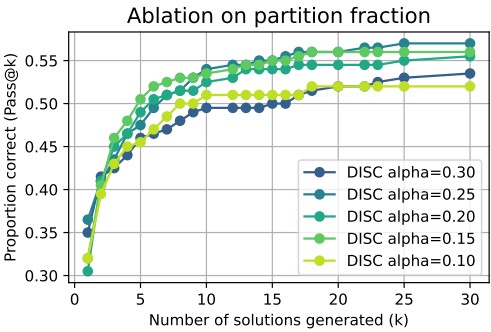

Figure 25: **Token level comparison of different DISC splitting fraction $\alpha_0$ on APPS competition level.** $0.15 \leq \alpha_0 \leq 0.25$ seems to be optimal.

Figure 26: **Effect of partition fraction $\alpha_0$ in DISC on APPS with gpt-4o-mini.** The range $0.15 \leq \alpha_0 \leq 0.25$ appears optimal.

We include additional analysis on the effect of the initial partition fraction $\alpha_0$, which determines the fraction of the best suffix used to propose a new candidate prefix at each iteration. As shown in Figures 25 and 26, we observe that performance peaks in the range $0.15 \leq \alpha_0 \leq 0.25$, with both token-level and Pass@k metrics favoring this region.

This behavior aligns with the underlying motivation of DISC: a smaller $\alpha_0$ results in more conservative candidate proposals—shorter steps that allow for finer-grained refinement. This is beneficial because committing to suboptimal prefixes early in the search can lead to irrevocable errors, especially in greedy search variants. Smaller values of $\alpha_0$ give the algorithm more flexibility to adjust course later on, while still making meaningful progress toward a complete solution. Conversely, large $\alpha_0$ values (e.g., $\alpha_0 > 0.3$) result in overly aggressive expansions that risk committing to noisy or premature completions, leading to reduced accuracy and inefficient use of sampling budget.

Therefore, the optimal range of $\alpha_0$ reflects a balance between exploration and commitment—proposing candidate steps that are informative enough to guide search, yet cautious enough to preserve the ability to refine future decisions. This ablation confirms that adaptive decomposition benefits from conservative, incremental prefix extension when navigating complex reasoning or program synthesis tasks.

## C.5 Reward Distribution

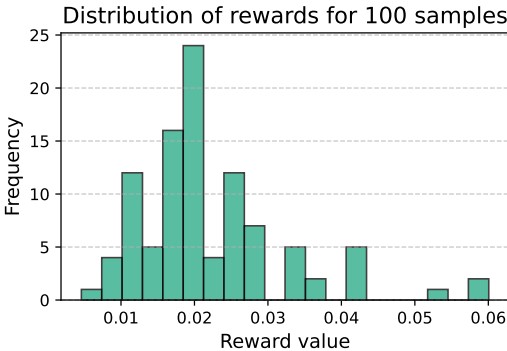

Figure 27: **Reward distribution for 100 samples on a given MATH500 problem.** The rewards appear to be roughly normal shaped.

A key assumption in our algorithm is that the distribution of rewards for sampled completions from a given prefix follows a location-scale family, such as the Gaussian distribution. This assumption enables the use of z-scores to estimate the relative quality of candidate prefixes and to guide step acceptance. As shown in Figure 27, the empirical reward distribution for 100 samples on a representative MATH500 problem appears approximately Gaussian, with a unimodal and symmetric shape. While we do not assume exact normality, this empirical observation supports the use of z-score-based comparisons, as the Gaussian approximation provides a reliable proxy for estimating tail probabilities and potential for reward improvement. We observe similar patterns across other problems and benchmarks, further validating this modeling choice.

# D Main Results Extended

## D.1 APPS

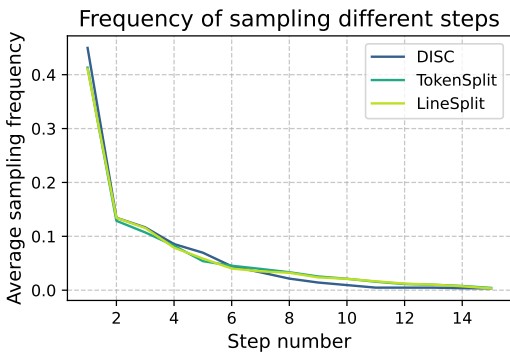

Figure 28: **Sampling frequency of each step averaged over the problems on APPS with gpt-4o-mini.** DISC seems to have a slight preferences for spending more compute on earlier found steps.

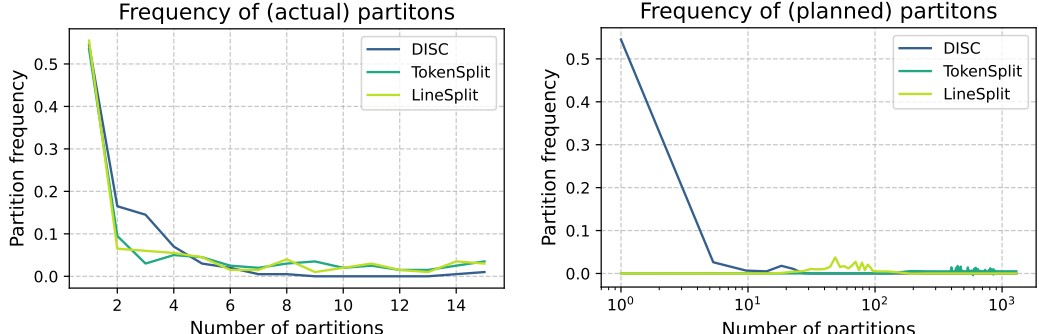

Figure 29: **Comparison of actual and planned partitions on APPS.** DISC outperforms other methods with fewer partitions by efficiently identifying critical steps. Unlike token and line split methods, which plan many partitions but search only a subset, DISC dynamically adjusts partitioning based on budget.

We provide extended analysis of DISC 's behavior on the APPS benchmark using `gpt-4o-mini`. Figure 28 shows the average sampling frequency per step across all problems. Interestingly, DISC exhibits a slight preference for allocating more sampling budget to earlier-discovered steps. This behavior reflects the intuition that early steps in a solution often set up the structure for later reasoning, and refining these foundational steps yields more downstream improvements.

Figure 29 compares the number of *actual* partitions (i.e., accepted and committed steps) with the number of *planned* partitions (i.e., proposed candidate splits) across decomposition methods. Token-level and line-based methods typically predefine many steps but only explore a subset due to budget constraints, leading to inefficient allocation. In contrast, DISC adaptively proposes and accepts partitions in real time based on sample statistics and reward feedback, resulting in significantly fewer actual partitions while maintaining or exceeding performance. This efficiency demonstrates DISC 's ability to identify and focus on high-impact steps, avoiding wasted compute on unnecessary refinements and enabling stronger performance within the same sampling budget.

### D.2 Additional Examples

Below is another computed decomposition for the same problem as the one shown in the main text:

---

**DISC example decomposition (Lagrange method)**

[ Let $x$ be the length and $y$ be the width of the rectangle. Since the perimeter is 24 inches, we have $2x + 2y = 24 \Rightarrow x + y = 12$. ] [ Therefore, ] [ we must maximize the area $xy$ subject to this constraint. ]

[ We use the method of Lagrange multipliers. The Lagrangian is

$$\mathcal{L}(x, y, \lambda) = xy - \lambda(x + y - 12).$$

We differentiate with respect to $x, y$, and $\lambda$ to obtain

$$\frac{\partial \mathcal{L}}{\partial x} = y - \lambda = 0, \qquad \frac{\partial \mathcal{L}}{\partial y} = x - \lambda = 0, \qquad \frac{\partial \mathcal{L}}{\partial \lambda} = x + y - 12 = 0.$$

We find that $x = y = \lambda$, so $x = y = \frac{12}{2} = 6$. Therefore, the maximum area of the rectangle is $6 \cdot 6 = \boxed{36}$. ]

---

DISC again highlights how initial constraint formulation is of medium importance—providing essential structure but not representing a reasoning fork. The token **"Therefore,"** is marked as high-importance, capturing a critical conceptual transition from constraint setup to the optimization approach. Notably, the bulk of the mathematical machinery involving partial derivatives and substitutions receives **low importance**, consistent with the idea that such computations are largely mechanical once the decision to use Lagrange multipliers is made. Interestingly, the final step involving the boxed answer is also marked low, suggesting DISC allocates minimal compute here once earlier reasoning is settled—underscoring the idea that key inference pivots lie upstream in the decision flow.

### D.3 APPS with Self-generated Validation Tests

We examine DISC performance on APPS when using self-generated validation tests. All methods utilized the same set of self-generated validation tests to ensure fair comparisons. Each problem received 5-10 validation tests, with the exact number determined dynamically by the LLM. We evaluated a subset of 100 APPS problems, generating samples until the sample budget was exhausted or a correct solution was found.

Figure 31 illustrates the Pass@token scaling curve, showing that DISC maintains strong scaling performance in this setting, though at a slightly lower rate compared to ground-truth verification.

Figure 32 and Figure 33 compare actual and planned partition frequencies, respectively. The results indicate that DISC continues to make structured decompositions even with self-generated validation, preserving its efficiency.

The mean reward per step, shown in Figure 34, follows a similar trend as in previous experiments, reinforcing that DISC effectively allocates compute resources for iterative refinement.

Lastly, Figure 35 demonstrates that DISC maintains lower standard deviations in performance, indicating stable quality improvements across steps.

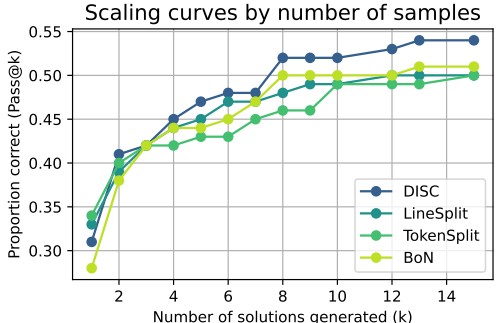

Figure 30: **Pass@k on APPS with gpt-4o-mini using self-generated validation tests.**

Figure 31: **Token level comparison of different decomposition methods on APPS with gpt-4o-mini and self-generated validation tests.** DISC still scales better than other methods in this setting, albeit at a lower rate.

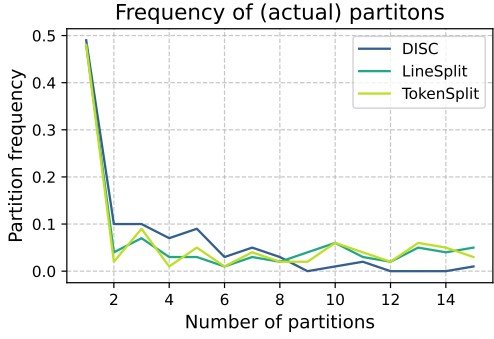

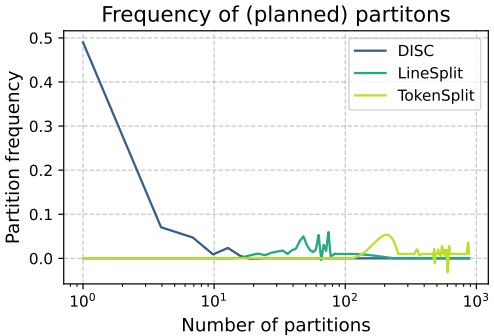

Figure 32: **Actual partition frequency of different decomposition methods on APPS with gpt-4o-mini and self-generated validation tests.**

Figure 33: **Planned partition frequency of different decomposition methods on APPS with gpt-4o-mini and self-generated validation tests.**

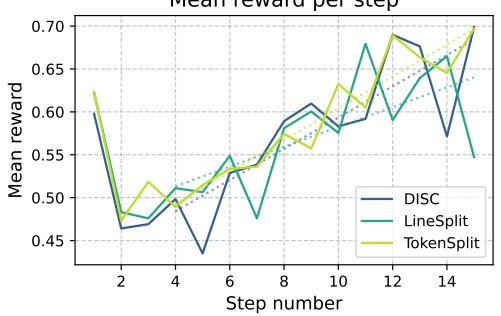

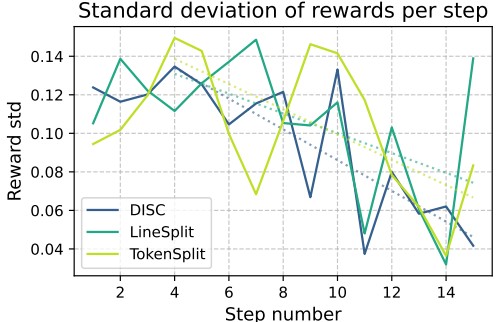

Figure 34: **Mean reward per step of different decomposition methods on APPS with gpt-4o-mini and self-generated validation tests.**

Figure 35: **Mean standard deviation different decomposition methods on APPS with gpt-4o-mini and self-generated validation tests.**

## D.4 MATH500

Completions for MATH500 include both the reasoning steps and the final answer. Since MATH500 contains more problems than APPS200 and MATH problems tend to be relatively easier, solution quality saturates quickly. Therefore, we use a lower sample budget of 10 for these experiments.

Figure 36 presents the Pass@k performance for different decomposition methods on MATH500. We observe that all decomposition-based approaches achieve similar Pass@k performance, consistently outperforming BoN. This indicates that the structured nature of MATH problems allows multiple decomposition strategies to be effective.

Despite similar Pass@k results, the true advantage of DISC lies in its token efficiency, as shown in Figure 5. DISC significantly reduces the number of tokens required to reach correct solutions compared to alternative methods, demonstrating its ability to allocate computational effort efficiently in mathematical reasoning tasks.

Additionally, we analyze the partitioning behavior of DISC on MATH500. Figure 37 illustrates the actual partition frequency for different decomposition methods. The planned partitioning behavior, shown in Figure 38, further highlights how DISC effectively balances exploration and refinement.

Finally, we present the mean standard deviation per step in Figure 39. Lower variance suggests that DISC produces more stable and reliable decompositions over multiple runs, reinforcing its robustness in both mathematical and program synthesis domains.

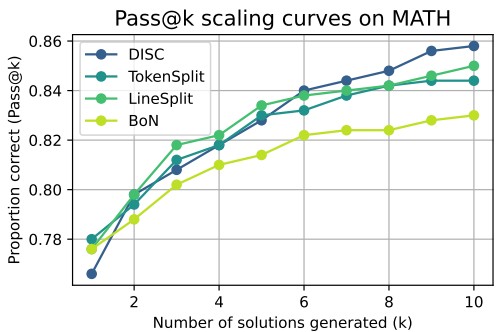

Figure 36: **Pass@k performance comparison for different decomposition methods on MATH500.** DISC consistently outperforms BoN across different sampling budgets.

Figure 37: **Observed partition frequency of different decomposition methods on MATH500.** DISC effectively segments problems into meaningful subcomponents.

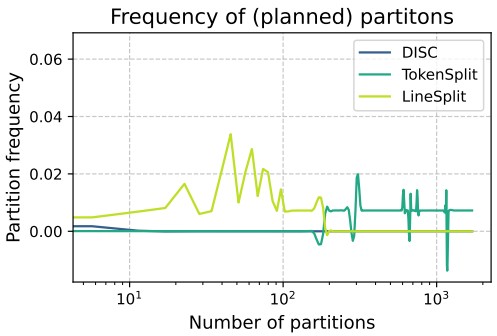

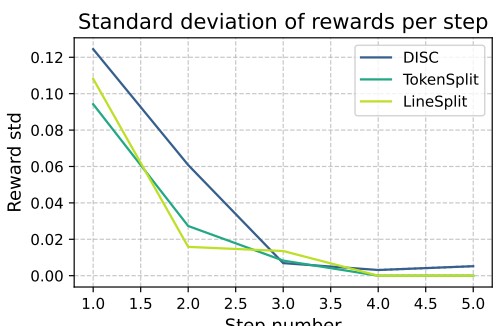

Figure 38: **Planned partitioning strategy of different decomposition methods on MATH500.** DISC 's structured approach leads to more efficient problem breakdowns.

Figure 39: **Mean standard deviation per step for different decomposition methods on MATH500.** Lower variance in DISC suggests more stable and reliable problem-solving steps.

## D.5    LiveCodeBench

We evaluate DISC on LiveCodeBench, a benchmark designed for code generation tasks with a focus on real-world software development challenges. LiveCodeBench presents a unique set of problems requiring both reasoning and structured decomposition, making it a suitable testbed for evaluating DISC's ability to refine and improve intermediate steps.

Figure 40 shows the Pass@k comparison of different decomposition methods on LiveCodeBench. DISC consistently scales better than other decomposition methods, highlighting its ability to refine intermediate steps more effectively in complex coding scenarios.

Figure 41 illustrates the observed partition frequency of different decomposition methods. The structured approach of DISC results in well-balanced decomposition across steps, reducing unnecessary partitioning while maintaining sufficient granularity for improved solution refinement.

Figure 42 displays the planned partition frequency across methods. DISC dynamically determines the most effective partitions based on the evolving problem state, leading to more targeted and efficient decompositions.

Finally, Figure 43 presents the mean standard deviation per step across decomposition methods. Lower variance in DISC suggests that it produces more stable and reliable decompositions, reinforcing its robustness for solving LiveCodeBench problems.

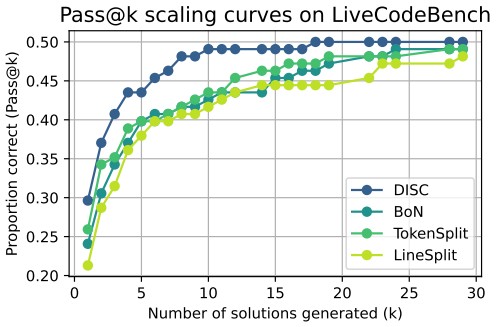

Figure 40: **Pass@k performance comparison for different decomposition methods on Live-CodeBench.** DISC consistently outperforms other methods in structured problem refinement.

Figure 41: **Observed partition frequency of different decomposition methods on Live-CodeBench.** DISC effectively balances problem segmentation while avoiding excessive partitioning.

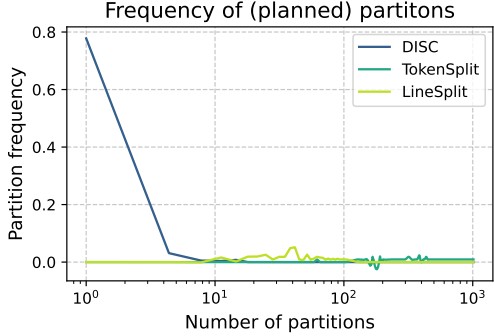

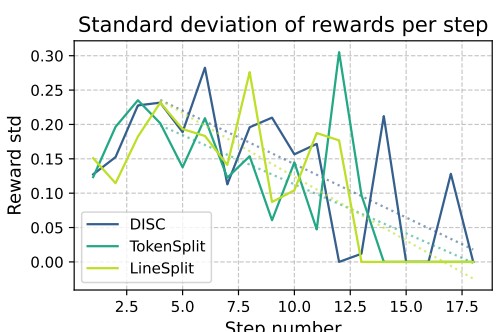

Figure 42: **Planned partitioning strategy of different decomposition methods on Live-CodeBench.** DISC dynamically adapts its partitioning to optimize search efficiency.

Figure 43: **Mean standard deviation per step for different decomposition methods on Live-CodeBench.** Lower variance in DISC suggests more stable and reliable problem-solving steps.

**D.6   Computational Overhead**

We analyze the computational overhead of DISC by measuring the runtime breakdown between actual LLM token generation and all auxiliary operations (e.g., z-score computation, candidate prefix management, and recursive decomposition logic). Since token generation is the dominant cost in LLM inference, reporting *relative percentages* of runtime—rather than absolute wall-clock time—offers a more meaningful and consistent comparison. Absolute runtime can vary significantly across hardware configurations, backend optimizations, or batching strategies, whereas percentage-based measurements isolate algorithmic overhead from infrastructure-dependent variance.

As shown in Figure 7, the proportion of runtime DISC spends on LLM token generation versus auxiliary logic closely matches that of baseline methods such as BoN and LineSplit. This indicates that DISC introduces negligible additional overhead despite its dynamic behavior and internal scoring computations. In practice, more than 90% of total runtime is still dominated by LLM token sampling across all methods.

Furthermore, because DISC achieves better performance with fewer tokens (i.e., improved token efficiency), its effective runtime per successful solution is actually *lower* than that of less efficient baselines. This combination—minimal added overhead and superior token scaling—makes DISC a highly practical method for real-world inference settings, where total compute cost is often tightly constrained.

# E   Limitations

While DISC demonstrates strong empirical performance and generality across tasks and models, several limitations merit discussion.

**Reliance on LLM-Generated Test Validation.**   Our evaluation framework employs *self-generated test validation*, where the LLM is prompted to produce its own unit tests for a given code generation task, and these tests are used as a proxy reward model. This approach enables scalable evaluation in the absence of ground-truth test cases, but its effectiveness depends critically on the quality of the generated tests. In particular:

- The LLM's ability to produce meaningful and comprehensive tests directly impacts evaluation fidelity. Poor coverage or semantically shallow tests may overestimate correctness.
- The framework assumes that most generated tests are correct and executable. For complex or underspecified tasks, this assumption may fail, leading to false positives or noisy reward signals.
- Since generated tests may not align with reference solutions, cross-method comparison can become inconsistent.

These issues are consistent with observations in prior work such as *CodeT* [30], which highlights tradeoffs between scalability and reliability in test-based self-evaluation.

**Dependence on Reward Model Availability.**   DISC requires access to a scalar reward signal to guide step-wise decomposition. While tasks like code generation or math reasoning naturally provide verifiers (e.g., test cases or numerical checks), applying DISC to domains lacking explicit reward signals necessitates constructing auxiliary reward models or LLM-based critics, which introduces additional complexity and potential bias.

**Scope Limited to Single-Turn Generation.**   The current formulation of DISC is designed for single-pass generation tasks. It does not yet handle multi-turn or interactive settings where intermediate reasoning steps can elicit feedback or modify the problem context. Extending DISC to such interactive regimes remains an important direction for future work.

**Reduced Benefit in Trivial or Non-Compositional Tasks.**   DISC allocates computational budget adaptively across reasoning steps. When a task is trivially solvable or lacks compositional structure—such that early reasoning does not meaningfully constrain the final outcome—the gains from decomposition and step-wise control diminish.

Despite these limitations, we believe that DISC provides a strong foundation for adaptive, reward-driven reasoning. Future extensions could integrate learned verifiers, support multi-turn interaction, and explore joint reasoning–evaluation co-evolution.

# F  Search and Scaling

## F.1  DISC Plug-and-Play Search

---

**Algorithm 3** DISC: Decomposition for Plug-and-Play Search

---

**Require:** LLM $\pi$, Reward model $R$, prompt $x$, initial partition fraction $\alpha_0$, negative binomial threshold $\sigma$, total budget $N$, current budget $n$

1: **function** EXPANDNODE(parent prefix $\boldsymbol{p}_b$, parent z score $z_b$, suffix to child $\boldsymbol{s}_c$)
2:     $\alpha = \alpha_0$
3:     **while** $n < N$ **do**
4:         $\boldsymbol{p}_c = \boldsymbol{p}_b \cdot \mathrm{split}(\boldsymbol{s}_c, \alpha)$
5:         $Y_c =$ MAKECHILDREN($\boldsymbol{p}_c$)
6:         $z_c = (\max(Y_c) - \mathrm{mean}(Y_c))/\mathrm{std}(Y_c)$
7:         $n \leftarrow n + M$
8:         **if** $z_c < z_b$ **then**
9:             **break**
10:        **else**
11:            $\alpha \leftarrow \alpha_0 \alpha$
12:        **end if**
13:    **end while**
14:    **return** $\boldsymbol{p}_c, Y_c$

15: **function** MAKECHILDREN(parent prefix $\boldsymbol{p}_b$)
16:     $Y = \{(\boldsymbol{p}_b \cdot \boldsymbol{s}^i, R(\boldsymbol{s}^i)) \mid \boldsymbol{s}^i \sim \pi(\cdot|\boldsymbol{p}_b)\}_{i=1}^{M}$
17:     **return** Y

---

## F.2  Monte Carlo Tree Search (MCTS)

Monte Carlo Tree Search (MCTS) is a widely used algorithm for sequential decision-making in large search spaces, particularly in applications such as *game playing, planning, and inference scaling*. The algorithm builds a search tree incrementally by simulating different sequences of actions and updating estimates of state quality. A key advantage of MCTS is its ability to balance *exploration* (discovering new states) and *exploitation* (refining promising ones) using a data-driven search process. The MCTS pipeline consists of four fundamental steps: *selection, expansion, simulation, and backpropagation*.

### F.2.1  Selection

Starting from the root node representing the current state $\boldsymbol{s}$, MCTS iteratively traverses the search tree by selecting child nodes based on a *selection policy*. The most commonly used selection criterion is the *Upper Confidence Bound for Trees (UCT)*, which balances exploration and exploitation:

$$UCT(\boldsymbol{s}, \boldsymbol{d}) = \hat{Q}(\boldsymbol{s}, \boldsymbol{d}) + c\sqrt{\frac{\ln\left(\sum_{\boldsymbol{b}} n(\boldsymbol{s}, \boldsymbol{b})\right)}{n(\boldsymbol{s}, \boldsymbol{d})}}, \tag{1}$$

where $\hat{Q}(\boldsymbol{s}, \boldsymbol{d})$ represents the estimated value of selecting action $\boldsymbol{d}$ from state $\boldsymbol{s}$, $n(\boldsymbol{s}, \boldsymbol{d})$ is the visit count for this action, and $c$ is a hyperparameter controlling the trade-off between exploring new actions and favoring those with high past rewards.

### F.2.2  Expansion

Once a leaf node (a previously unexplored state) is reached, the algorithm expands the tree by *adding one or more new nodes*. These new nodes represent potential future states $\boldsymbol{s}'$ generated by sampling an action $\boldsymbol{d}$ from a predefined policy. This step broadens the search space and allows MCTS to evaluate new possibilities.

### F.2.3 Simulation

Following expansion, the algorithm conducts a *simulation* (or rollout) from the newly added state. This step involves generating a sequence of actions according to a predefined policy until reaching a terminal state or an evaluation horizon. The outcome of the simulation, denoted as $v(s')$, provides an estimate of the quality of the new state. Depending on the application, this could represent a *game result, an optimization score, or an inference accuracy metric*.

### F.2.4 Backpropagation

The final step involves *propagating the results of the simulation back up the search tree* to refine the estimated values of prior states and actions. Each node along the trajectory $\tau = [s_0, d_1, s_2, \ldots, s_{-1}]$ is updated iteratively:

$$\hat{Q}(s_i, d_{i+1})^{(t+1)} \leftarrow (1 - \alpha_n)\hat{Q}(s_i, d_{i+1})^{(t)} + \alpha_n \max\{\hat{Q}(s_i, d_{i+1})^{(t)}, \hat{Q}(s_{i+1}, d_{i+2})^{(t+1)}\}, \quad (2)$$

where $\alpha_n$ is a learning rate that depends on the visit count, and the maximum function ensures that the best-performing trajectories are emphasized.

MCTS has been widely adopted in inference scaling techniques due to its ability to *efficiently allocate computational resources*, focusing more on *high-reward states* while avoiding unnecessary exploration of unpromising regions. In later sections, we explore how MCTS can be combined with *dynamic decomposition* to further optimize inference scaling.

### F.2.5 Combining Dynamic Decomposition with MCTS

MCTS can be enhanced by integrating *dynamic decomposition*, where each node in the search tree represents a decomposition of the problem into steps. Instead of treating states as atomic decisions, we recursively decompose reasoning steps, dynamically adjusting granularity based on difficulty.

In this framework:

- Each node in the MCTS tree represents a partial decomposition of the problem, with child nodes corresponding to alternative step partitions.
- Branching occurs by generating candidate next steps using dynamic decomposition, allowing finer steps for complex regions while maintaining efficiency for simpler ones.
- The selection step prioritizes nodes that represent more promising decompositions, dynamically refining challenging areas through recursive subdivision.
- The backpropagation step ensures that decompositions leading to high-quality solutions are reinforced, helping the search tree converge toward optimal inference paths.

By integrating dynamic decomposition with MCTS, we efficiently allocate compute to the most critical reasoning steps, improving inference quality while maintaining computational efficiency.

### F.3 Beam Search

Beam search is a heuristic search algorithm commonly used in inference tasks where computational efficiency is a priority. Unlike exhaustive search methods, beam search maintains only the top $k$ best candidates at each step, making it an effective strategy for structured prediction problems and sequential decision-making.

At each iteration:

- The algorithm selects the $k$ most promising partitions from the previous step based on an evaluation metric.
- Each selected partition is expanded by generating possible next-step samples.
- The newly generated partitions are ranked, and only the top $k$ candidates are retained for the next iteration.
- This process continues until a stopping criterion is met, such as reaching a predefined depth or finding a sufficiently high-quality solution.

Beam search provides a computationally efficient way to explore structured solution spaces while maintaining high-quality search trajectories. By integrating beam search with dynamic decomposition, we ensure that inference computation is allocated efficiently, focusing on the most promising reasoning paths at each step.

## F.4   Additional Results and Analysis

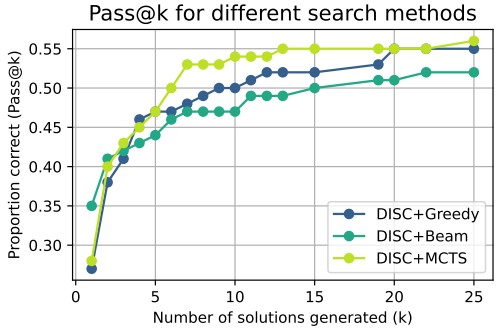

Figure 44: **Pass@k on APPS with gpt-4o-mini using different search methods.** MCTS scales best, followed by greedy search, followed by beam search.

Figure 45: **Token level comparison of different decomposition search methods combined with DISC on APPS with gpt-4o-mini.** MCTS scales best, followed by greedy search, followed by beam search.

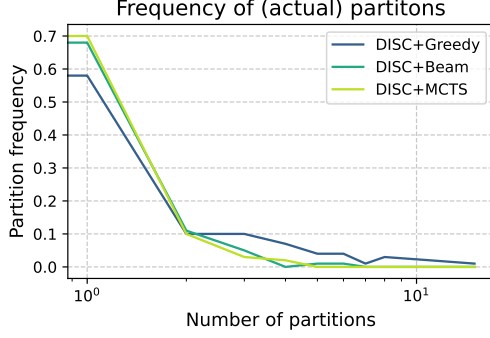

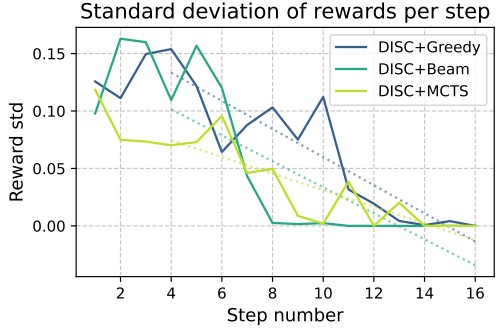

Figure 46: **Actual partition frequency of different decomposition search methods combined with DISC on APPS with gpt-4o-mini.** Greedy is able to search to higher depths given the same sampling budget.

Figure 47: **Mean standard deviation of different decomposition search methods combined with DISC on APPS with gpt-4o-mini.** All search methods display decreasing standard deviation with search depth.

Experiments comparing different search methods were conducted on a 100-problem subset of the APPS dataset (first 100 problems) using GPT-4o-mini. All methods used a temperature of 0.2, with $\alpha = 0.15$, Q priority metric, and $\sigma = 1.0$.

**Token-level comparison:** As shown in Figure 45, MCTS scales best among the tested methods, demonstrating superior efficiency in identifying promising partitions. Greedy search follows closely, while beam search exhibits the slowest scaling.

**Partition frequency analysis:** Figure 46 reveals that greedy search explores to greater depths within the same sampling budget. This suggests that greedy search prioritizes deep refinements, whereas MCTS and beam search balance depth with breadth.

**Step variance analysis:** Figure 47 illustrates that all search methods display decreasing standard deviation with increasing search depth. This trend indicates that deeper searches converge towards stable, high-quality partitions, reinforcing the benefits of dynamic decomposition.

These results highlight the trade-offs between search methods: MCTS offers robust exploration-exploitation balance, greedy search favors depth-first refinement, and beam search provides a structured yet computationally constrained approach. The integration of dynamic decomposition further enhances these search strategies by adaptively allocating computational resources to critical reasoning steps.

# G Theoretical analysis

## G.1 Main Theorem

We begin by introducing notation to characterize the distribution of rewards obtained when sampling completions from a given prefix $\boldsymbol{p}$.

Let $R_{\boldsymbol{p}}$ denote the random variable of the reward of a completion sampled from the LLM policy $\pi$ conditioned on prefix $\boldsymbol{p}$:

$$R_{\boldsymbol{p}} := R(\boldsymbol{p} \cdot \boldsymbol{s}), \quad \text{where } \boldsymbol{s} \sim \pi(\cdot \mid \boldsymbol{p}).$$

Let $F_{\boldsymbol{p}}$, $\mu_{\boldsymbol{p}}$, and $\sigma_{\boldsymbol{p}}$ be the cumulative distribution function (CDF), mean, and standard deviation of $R_{\boldsymbol{p}}$, respectively.

We define a solution $\boldsymbol{y}^*$ as *correct* if it achieves a reward of at least $r^* = 1$, i.e., $R(\boldsymbol{y}^*) \geq 1$. Then, the probability that a random sample from $\pi(\cdot \mid \boldsymbol{p})$ yields a correct solution is given by:

$$\mathbb{P}[R_{\boldsymbol{p}} \geq 1] = 1 - F_{\boldsymbol{p}}(1).$$

For a given set of $M$ independent samples from $\pi(\cdot \mid \boldsymbol{p})$, let $r_{\boldsymbol{p}}^{\max}$ denote the maximum observed reward among them, i.e., the sample maximum.

We make the following problem assumption:

**Assumption 1** *A correct solution exists in the support of the policy $\pi(\boldsymbol{y}^* \mid \boldsymbol{x}) > 0$ and the length of the correct solution is finite: $|\boldsymbol{y}^*| = K < \infty$.*

We make the following assumptions about the policy $\pi$:

**Assumption 2** *We assume $R_{\boldsymbol{p}}$ belongs to a location-scale family with base CDF $F_R$.*

**Remark 2** *The location-scale family includes common distributions such as the Normal, Cauchy, and Logistic distributions, which are characterized by being affine transformations of a fixed base distribution. Specifically, a random variable $X$ belongs to a location-scale family if it can be written as $X = \mu + \sigma Z$, where $Z \sim F_R$ is a standardized random variable, and $\mu \in \mathbb{R}$ and $\sigma > 0$ are the location and scale parameters, respectively. We observe empirical evidence supporting this modeling choice for $R_{\boldsymbol{p}}$ in Appendix C.5.*

**Assumption 3 (Reward distribution converges with prefix)** *Consider some base prefix $\boldsymbol{b}$ and some candidate prefix $\boldsymbol{c} = \boldsymbol{b} \cdot \boldsymbol{x}$ for any token sequence $\boldsymbol{x}$. Then, $\sigma_{\boldsymbol{c}} \leq \sigma_{\boldsymbol{b}}$. Furthermore, if $\boldsymbol{b}$ is a complete solution (it ends in an EOS token) then $\sigma_{\boldsymbol{b}} = 0$.*

**Remark 3** *This assumption reflects the intuition that as more tokens are appended to a prefix, the LLM becomes increasingly committed to a narrower set of likely continuations, leading to lower uncertainty in the reward distribution. In the limit, once a complete solution is formed (i.e., the sequence ends with an EOS token), the reward becomes deterministic, and the variance collapses to zero.*

*We emphasize that this assumption is not required for proving the optimality of DISC. However, the greater the difference between $\sigma_{\boldsymbol{c}} < \sigma_{\boldsymbol{b}}$, the faster the convergence, as DISC will be able to prune suboptimal paths more confidently with fewer samples.*

**Assumption 4 (Reward distribution converges slowly)** *If Algorithm 1 accepts a candidate prefix $\boldsymbol{c}$ over a base prefix $\boldsymbol{b}$, then we assume the standard deviation of the reward distribution under $\boldsymbol{c}$ is a lower bounded:*

$$\sigma_{\boldsymbol{c}} \geq \left(1 - \frac{\delta \sigma_{\boldsymbol{c}}}{\Delta}\right) \sigma_{\boldsymbol{b}}$$

*where $\Delta = r^* - r_{\boldsymbol{c}}^{\max}$ is the **optimality gap** between the true maximum reward $r^*$ and the current best sample under prefix $\boldsymbol{b}$, and $\delta = \frac{r_{\boldsymbol{b}}^{\max} - \mu_{\boldsymbol{b}}}{\sigma_{\boldsymbol{b}}} - \frac{r_{\boldsymbol{c}}^{\max} - \mu_{\boldsymbol{c}}}{\sigma_{\boldsymbol{c}}}$ is the **z-score drop**, which quantifies the change in how surprising the best sample is under each distribution.*

**Remark 4** *This assumption enforces that if we transition from base prefix $\boldsymbol{b}$ to candidate prefix $\boldsymbol{c}$, the standard deviation (spread) of the reward distribution under $\boldsymbol{c}$ cannot shrink too drastically—especially when: (i) we are still far from the optimal reward (i.e., $\Delta$ is large), and (ii) the improvement in sample quality is not significant (i.e., $\delta$ is small). Intuitively, this prevents committing to a sharp but unreliable prefix unless there is strong evidence of progress. If $\boldsymbol{c}$ does not yield sufficiently better samples or has low variance, it may hinder further exploration. This assumption ensures that accepted prefixes preserve enough reward diversity to support continued search.*

**Remark 5** *While we list the confidence property Assumption 4 as an assumption, it can also be enforced directly as an **explicit acceptance criterion** in Algorithm 1. Specifically, the algorithm could be modified to accept a candidate prefix $\boldsymbol{c}$ over base $\boldsymbol{b}$ only if both $\delta \geq 0$ and: $\left(1 - \frac{\delta \sigma_{\boldsymbol{c}}}{\Delta}\right) \sigma_{\boldsymbol{b}} \leq \sigma_{\boldsymbol{c}}$. where $\Delta$ can be computed because $r^* = 1$.*

Combining Assumptions 3 and 4, we can write: $\frac{\sigma_c}{\sigma_b} \in \left[\left(1 - \frac{\delta \sigma_c}{\Delta}\right), 1\right]$. This balances exploration with exploitation.

**Assumption 5 (Accurate estimates of $\mu_{\boldsymbol{p}}$ and $\sigma_{\boldsymbol{p}}$ )** *We assume that we sample $R_{\boldsymbol{p}}$ enough times to get accurate estimates of the true mean $\hat{\mu}_{\boldsymbol{p}} \approx \mu_{\boldsymbol{p}}$ and standard deviation $\hat{\sigma}_{\boldsymbol{p}} \approx \sigma_{\boldsymbol{p}}$. Because estimates of mean and standard deviation are accurate, the estimates of the z-scores are also accurate.*

Next we prove that Alg. 1 will never commit a candidate prefix that with a lower probability of sampling a correct solution than the base.

**Lemma 1** *Consider Algorithm 1, applied to a base prefix $\boldsymbol{b}$ and a candidate prefix $\boldsymbol{c}$, with corresponding reward distributions $F_{\boldsymbol{b}}$ and $F_{\boldsymbol{c}}$. Then, the probability of sampling a correct suffix from $\pi$ does not decrease after accepting $\boldsymbol{c}$:*

$$\pi(\boldsymbol{s}^* \mid \boldsymbol{b}) \leq \pi(\boldsymbol{s}^* \mid \boldsymbol{c}).$$

**Proof 1** *A candidate prefix $\boldsymbol{c}$ is accepted by Algorithm 1 if either of the two conditions are true: (i) the standardized score (z-score) of $\boldsymbol{c}$ is lower than that of $\boldsymbol{b}$: $z_c < z_b$ or (ii) the first $\alpha$ fraction of tokens in the best suffix contains no tokens, and the prefix is accepted by default. In case (ii): if $\boldsymbol{c}$ is the empty prefix, then $\boldsymbol{c} = \boldsymbol{b}$, and the distributions are unchanged, so the result follows trivially.*

*It remains to show case (i): if $z_c < z_b$ then, $\pi(\boldsymbol{s}^* \mid \boldsymbol{c}) \geq \pi(\boldsymbol{s}^* \mid \boldsymbol{b})$.*

*First, we define the z-score change $\delta > 0$ as $\delta = \frac{r_b^{\max} - \mu_{\boldsymbol{b}}}{\sigma_{\boldsymbol{b}}} - \frac{r_c^{\max} - \mu_{\boldsymbol{c}}}{\sigma_{\boldsymbol{c}}}$ and the optimality gap $\Delta > 0$ as $\Delta = r^* - r_{\boldsymbol{c}}^{\max}$. Then, we note that our algorithmic implementation initializes the set of candidate samples with the best solution, i.e. $\max(Y_b) \in Y_c$. Therefore, we have that the sample max of the candidate rewards is greater than or equal to that of the base: $r_{\boldsymbol{c}}^{\max} \geq r_{\boldsymbol{b}}^{\max}$.*

*Using this information, we can rewrite the z-score inequality:*

$$\frac{r^* - \mu_{\boldsymbol{b}}}{\sigma_{\boldsymbol{b}}} - \frac{r^* - \mu_{\boldsymbol{c}}}{\sigma_{\boldsymbol{c}}} \geq \left(\delta - \frac{\Delta}{\sigma_{\boldsymbol{c}}} + \frac{\Delta}{\sigma_{\boldsymbol{b}}}\right) \geq 0 \tag{3}$$

*where we first plug in the definitions of $\delta$ and $\Delta$, do algebraic manipulation, and then apply assumption 4. The final inequality implies $F_{\boldsymbol{c}}(r^*) < F_{\boldsymbol{b}}(r^*)$, which implies $\pi(\boldsymbol{s}^* \mid \boldsymbol{c}) \geq \pi(\boldsymbol{s}^* \mid \boldsymbol{b})$ because $\mathbb{P}[R_{\boldsymbol{p}} \geq r^*] = 1 - F_{\boldsymbol{p}}(r^*)$ and $F_{\boldsymbol{p}}$ is monotonic in z score.*

**Theorem 2 (Optimality of DISC)** *Consider Algorithm 1 applied to a problem input $\boldsymbol{x}$ and assume Assumptions 1, 2, 4, and 5 hold. Then, with probability 1, there exists a finite number of accepted prefixes $k > 0$ and algorithm iterations $n > 0$ such that the algorithm terminates, and the best solution $\boldsymbol{y}_k$ found is a correct solution: $R(\boldsymbol{y}_k) \geq 1$.*

**Proof 2** *We define the base prefix $\boldsymbol{b}_k$ as the result of appending $k$ accepted candidate prefixes. We now proceed with the induction.*

*We first prove by induction that the probability of sampling a correct suffix remains strictly positive after each accepted prefix.*

***Base Case** ($k = 0$): Initially, $\boldsymbol{b}_0 = \boldsymbol{x}$. By assumption, there exists at least one correct solution $\boldsymbol{y}^* = \boldsymbol{x} \cdot \boldsymbol{s}^*$ such that $\pi(\boldsymbol{s}^* \mid \boldsymbol{x}) > 0$ and $R(\boldsymbol{y}^*) \geq 1$. Hence, there is a non-zero probability of sampling a correct suffix from $\boldsymbol{b}_0$.*

***Inductive Step:*** *Assume that after $k$ accepted prefixes, the base prefix $\boldsymbol{b}_k$ satisfies:*

$$\pi(\boldsymbol{s}^* \mid \boldsymbol{b}_k) > 0 \quad and \quad R(\boldsymbol{b}_k \cdot \boldsymbol{s}^*) \geq 1$$

*for some correct suffix $\boldsymbol{s}^*$. Let $\boldsymbol{b}_{k+1} = \boldsymbol{b}_k \cdot \boldsymbol{c}$ be the base prefix after appending the $(k+1)$-th accepted candidate prefix $\boldsymbol{c}$. By Lemma 1, the probability of sampling a correct suffix does not decrease:*

$$\pi(\boldsymbol{s}^* \mid \boldsymbol{b}_{k+1}) \geq \pi(\boldsymbol{s}^* \mid \boldsymbol{b}_k) > 0.$$

*Therefore, at every accepted prefix $\boldsymbol{b}_k$, the probability of sampling a correct suffix remains bounded below by some $\varepsilon > 0$, where $\varepsilon \geq \pi(\boldsymbol{y}^* \mid \boldsymbol{x})$.*

***Termination:*** *Let $M_k$ be the number of samples from accepted candidate prefix $\boldsymbol{b}_k$. Since we sample from the accepted candidate prefix $\boldsymbol{b}_k$ at least once ($M_k \geq 1$), samples are independent, and the probability of sampling a correct suffix is at least $\pi(\boldsymbol{y}^* \mid \boldsymbol{x}) > 0$, the probability of not sampling a correct solution after accepting $k$ prefixes is at most*

$$\left(1 - \pi(\boldsymbol{y}^* \mid \boldsymbol{x})\right)^k.$$

*Hence, the probability of never sampling a correct solution after infinitely many accepted prefixes is*

$$\lim_{k \to \infty} \left(1 - \pi(\boldsymbol{y}^* \mid \boldsymbol{x})\right)^k = 0,$$

*which implies that, with probability 1, the algorithm will eventually sample a correct suffix.*

*Furthermore, this implies that the number of accepted prefixes $k$ before obtaining a correct solution is almost surely finite. This follows from the fact that the number of independent trials before the first success with fixed success probability $p := \pi(\boldsymbol{y}^* \mid \boldsymbol{x}) > 0$ is a geometric random variable, which is finite with probability 1.*

*Therefore, there exists a finite number of accepted prefixes $k > 0$ such that the algorithm terminates and returns a correct solution $\boldsymbol{y}^*$ satisfying $R(\boldsymbol{y}^*) \geq 1$.*

*Since the algorithm either accepts a candidate prefix or contracts it by a factor $\alpha \in (0, 1)$ until the prefix length is 0, there can only be finitely many contractions before the candidate prefix is rejected or accepted. Therefore the algorithm must accept a candidate prefix in finite number of iterations, and so the total number of algorithm iterations $n$ before finding a correct solution is also finite.*

**Remark 6** *Unlike Best-of-N (BoN), where the probability of sampling a correct solution remains constant across samples, DISC dynamically refines the sampling distribution by appending informative prefixes. As established in Lemma 1, the probability of generating a correct solution under DISC is non-decreasing across iterations. Consequently, DISC achieves faster convergence to a correct solution in expectation compared to BoN. This theoretical advantage is corroborated by our empirical results in Figure 5, which show that DISC consistently outperforms BoN in terms of efficiency and scalability with respect to token budget.*

## G.2 A Motivating Example on DISC

We use the Wiener process $W(t)$ as an example where there are intractably many actions and steps. Suppose we start at $t = 0$ with $W(0) = 0$. At each round $k$, the algorithm can choose one of the two options:

1. samples a trajectory and observe the final value $W(T)$ at time $t = T$, as the reward signal. Denote the whole trajectory as $w_k(\cdot)$.
2. chooses one trajectory from the previous rounds (denoted as $w_s(t)$ for round $s$), and time $t_0$; then sample a trajectory at $t = t_0$ with $W(t_0) = w_s(t_0)$. Denote the concatenated trajectory as $w_k(\cdot)$ with $w_k(t) = w_s(t)$ when $t \leq t_0$.

Note that we are only able to observe the final reward $W(t)$. At any intermediate time $t \in (0, T)$, the current value $W(t)$ is not observable. The goal is to design an algorithm that can reach the highest reward among the $K$ trajectories. Formally speaking, we aim to maximize the maximum:

$$\max_{k \in K} w_k(T).$$

One naive solution is to call option 1 for $K$ times and return the best-of-$K$ reward, each following:

$$W(T) \sim \mathcal{N}(0, T).$$

Alternatively, suppose there is a promising path $w(\cdot)$ with a high final reward $w(T) = R$. It is natural to consider starting at some midpoint $\alpha T$ ($0 < \alpha < 1$) and perform more completions to obtain an even higher reward than $R$. The reward distribution sampled this way is

$$W'(T) \sim \mathcal{N}(w(\alpha T), (1 - \alpha)T).$$

The remaining question is which $\alpha$ we should choose. One option is to maximize the probability that the newly sampled reward is higher than $R$:

$$\mathbb{P}(W'(T) > R) = 1 - \Phi\left(\frac{R - w(\alpha T)}{\sqrt{(1 - \alpha)T}}\right).$$

# H  Compute Resources Used

All evaluations involving OpenAI proprietary models (e.g., GPT-4o-mini, GPT-3.5) were performed via the OpenAI API. These API calls were made from standard desktop machines using CPU-only inference and incur no dependency on local GPU availability, making the method broadly accessible for replication.

For open-source model experiments (e.g., LLaMA-3.1-8B-Instruct, Mistral, Qwen), inference was conducted on a single NVIDIA A100 GPU. All such experiments were executed sequentially on this GPU, which has 80GB of memory, ensuring consistent and reproducible runtime characteristics across model families.

This setup reflects the lightweight computational overhead of our method and demonstrates that DISC can scale effectively even in constrained or CPU-only environments when using hosted APIs.

# I  Impact Statement

**Positive Societal Impacts**   This work introduces a general and lightweight method—Dynamic Decomposition (DISC)—that significantly improves inference efficiency for large language models (LLMs) without requiring additional training, domain-specific engineering, or specialized hardware. By enabling better performance using fewer samples and tokens, DISC reduces the cost and environmental footprint associated with LLM deployment, making advanced reasoning models more accessible to research groups and developers with limited compute budgets.

Additionally, DISC's ability to prioritize critical reasoning steps has the potential to improve the transparency and interpretability of LLM outputs, which is beneficial in high-stakes applications such as education, scientific reasoning, and assistive tools for programming or mathematics. Its plug-and-play compatibility with open-source models also democratizes access to high-performance inference techniques.

**Negative Societal Impacts**   As with all improvements in LLM inference capabilities, this work may accelerate the deployment of models in settings where societal risks are not fully mitigated—such as the automated generation of persuasive misinformation, cheating in educational contexts, or manipulation via high-fidelity language generation. Furthermore, by making LLM inference more efficient, DISC could contribute to increased usage of models without corresponding increases in ethical oversight or alignment safeguards.

Careful integration of this method should therefore include responsible use policies, limitations on deployment domains, and alignment with values such as transparency, fairness, and accountability.

# NeurIPS Paper Checklist

The checklist is designed to encourage best practices for responsible machine learning research, addressing issues of reproducibility, transparency, research ethics, and societal impact. Do not remove the checklist: **The papers not including the checklist will be desk rejected.** The checklist should follow the references and follow the (optional) supplemental material. The checklist does NOT count towards the page limit.

Please read the checklist guidelines carefully for information on how to answer these questions. For each question in the checklist:

- You should answer [Yes] , [No] , or [NA] .
- [NA] means either that the question is Not Applicable for that particular paper or the relevant information is Not Available.
- Please provide a short (1–2 sentence) justification right after your answer (even for NA).

**The checklist answers are an integral part of your paper submission.** They are visible to the reviewers, area chairs, senior area chairs, and ethics reviewers. You will be asked to also include it (after eventual revisions) with the final version of your paper, and its final version will be published with the paper.

The reviewers of your paper will be asked to use the checklist as one of the factors in their evaluation. While "[Yes] " is generally preferable to "[No] ", it is perfectly acceptable to answer "[No] " provided a proper justification is given (e.g., "error bars are not reported because it would be too computationally expensive" or "we were unable to find the license for the dataset we used"). In general, answering "[No] " or "[NA] " is not grounds for rejection. While the questions are phrased in a binary way, we acknowledge that the true answer is often more nuanced, so please just use your best judgment and write a justification to elaborate. All supporting evidence can appear either in the main paper or the supplemental material, provided in appendix. If you answer [Yes] to a question, in the justification please point to the section(s) where related material for the question can be found.

IMPORTANT, please:

- **Delete this instruction block, but keep the section heading "NeurIPS Paper Checklist"**,
- **Keep the checklist subsection headings, questions/answers and guidelines below.**
- **Do not modify the questions and only use the provided macros for your answers**.

1. **Claims**

   Question: Do the main claims made in the abstract and introduction accurately reflect the paper's contributions and scope?

   Answer: [Yes]

   Justification: The abstract and introduction accurately reflect the paper's contributions and scope. The paper introduces a novel framework for decomposing behavior in sequential decision-making using value-based segmentation, proposes an instantiation via a modified value iteration algorithm, and demonstrates empirical results across diverse domains including control, navigation, and language modeling. These contributions are clearly outlined in both the abstract and introduction and are elaborated upon in Sections 2 (Framework), 3 (Method), and 4 (Experiments).

   Guidelines:
   - The answer NA means that the abstract and introduction do not include the claims made in the paper.
   - The abstract and/or introduction should clearly state the claims made, including the contributions made in the paper and important assumptions and limitations. A No or NA answer to this question will not be perceived well by the reviewers.
   - The claims made should match theoretical and experimental results, and reflect how much the results can be expected to generalize to other settings.
   - It is fine to include aspirational goals as motivation as long as it is clear that these goals are not attained by the paper.

2. **Limitations**

   Question: Does the paper discuss the limitations of the work performed by the authors?

   Answer: [Yes]

   Justification: The limitations are discussed in Section 3.1, where the authors acknowledge that the value-based decomposition relies on having access to a value function, which may not always be readily available or accurate in real-world settings. They also note that while the method is general, its effectiveness can depend on the specific domain and value structure. These points transparently address the boundaries of the current approach.

   Guidelines:

   - The answer NA means that the paper has no limitation while the answer No means that the paper has limitations, but those are not discussed in the paper.
   - The authors are encouraged to create a separate "Limitations" section in their paper.
   - The paper should point out any strong assumptions and how robust the results are to violations of these assumptions (e.g., independence assumptions, noiseless settings, model well-specification, asymptotic approximations only holding locally). The authors should reflect on how these assumptions might be violated in practice and what the implications would be.
   - The authors should reflect on the scope of the claims made, e.g., if the approach was only tested on a few datasets or with a few runs. In general, empirical results often depend on implicit assumptions, which should be articulated.
   - The authors should reflect on the factors that influence the performance of the approach. For example, a facial recognition algorithm may perform poorly when image resolution is low or images are taken in low lighting. Or a speech-to-text system might not be used reliably to provide closed captions for online lectures because it fails to handle technical jargon.
   - The authors should discuss the computational efficiency of the proposed algorithms and how they scale with dataset size.
   - If applicable, the authors should discuss possible limitations of their approach to address problems of privacy and fairness.
   - While the authors might fear that complete honesty about limitations might be used by reviewers as grounds for rejection, a worse outcome might be that reviewers discover limitations that aren't acknowledged in the paper. The authors should use their best judgment and recognize that individual actions in favor of transparency play an important role in developing norms that preserve the integrity of the community. Reviewers will be specifically instructed to not penalize honesty concerning limitations.

3. **Theory assumptions and proofs**

   Question: For each theoretical result, does the paper provide the full set of assumptions and a complete (and correct) proof?

   Answer: [Yes]

   Justification: The paper presents a clear set of theoretical results in Section 3, supported by assumptions which are stated explicitly. Lemma 1 and Theorem 1 build directly on these assumptions and are accompanied by informal but complete proofs that are consistent with the problem setting. The paper also explicitly states the limitations of its theoretical analysis, acknowledging that some assumptions are empirically motivated rather than rigorously provable, which is standard in empirical LLM research. The justification is sufficient and appropriate for a NeurIPS submission.

   Guidelines:

   - The answer NA means that the paper does not include theoretical results.
   - All the theorems, formulas, and proofs in the paper should be numbered and cross-referenced.
   - All assumptions should be clearly stated or referenced in the statement of any theorems.
   - The proofs can either appear in the main paper or the supplemental material, but if they appear in the supplemental material, the authors are encouraged to provide a short proof sketch to provide intuition.

- Inversely, any informal proof provided in the core of the paper should be complemented by formal proofs provided in appendix or supplemental material.
- Theorems and Lemmas that the proof relies upon should be properly referenced.

4. **Experimental result reproducibility**

Question: Does the paper fully disclose all the information needed to reproduce the main experimental results of the paper to the extent that it affects the main claims and/or conclusions of the paper (regardless of whether the code and data are provided or not)?

Answer: [Yes]

Justification: The paper provides detailed descriptions of the experimental setup, including datasets (APPS, MATH500, and LiveCodeBench), evaluation metrics (pass@k, pass@token), model configurations (e.g., LLaMA-3.1-8B-Instruct, GPT-4o-mini), sampling temperature ($\tau = 0.2$), and DISC hyperparameters (e.g., $\epsilon_0 = 0.15, \theta = 1.0$). The algorithm is specified in pseudocode (Alg. 1 and 2), and ablation studies are conducted to analyze sensitivity to temperature, acceptance criteria, and partition fraction. While code is not included in the paper, the level of detail is sufficient to reimplement the approach and verify the core findings.

Guidelines:

- The answer NA means that the paper does not include experiments.
- If the paper includes experiments, a No answer to this question will not be perceived well by the reviewers: Making the paper reproducible is important, regardless of whether the code and data are provided or not.
- If the contribution is a dataset and/or model, the authors should describe the steps taken to make their results reproducible or verifiable.
- Depending on the contribution, reproducibility can be accomplished in various ways. For example, if the contribution is a novel architecture, describing the architecture fully might suffice, or if the contribution is a specific model and empirical evaluation, it may be necessary to either make it possible for others to replicate the model with the same dataset, or provide access to the model. In general. releasing code and data is often one good way to accomplish this, but reproducibility can also be provided via detailed instructions for how to replicate the results, access to a hosted model (e.g., in the case of a large language model), releasing of a model checkpoint, or other means that are appropriate to the research performed.
- While NeurIPS does not require releasing code, the conference does require all submissions to provide some reasonable avenue for reproducibility, which may depend on the nature of the contribution. For example
  (a) If the contribution is primarily a new algorithm, the paper should make it clear how to reproduce that algorithm.
  (b) If the contribution is primarily a new model architecture, the paper should describe the architecture clearly and fully.
  (c) If the contribution is a new model (e.g., a large language model), then there should either be a way to access this model for reproducing the results or a way to reproduce the model (e.g., with an open-source dataset or instructions for how to construct the dataset).
  (d) We recognize that reproducibility may be tricky in some cases, in which case authors are welcome to describe the particular way they provide for reproducibility. In the case of closed-source models, it may be that access to the model is limited in some way (e.g., to registered users), but it should be possible for other researchers to have some path to reproducing or verifying the results.

5. **Open access to data and code**

Question: Does the paper provide open access to the data and code, with sufficient instructions to faithfully reproduce the main experimental results, as described in supplemental material?

Answer: [No]

Justification: The code and data are not yet publicly available at submission time due to anonymity constraints. However, we commit to releasing the full implementation—including scripts for running DISC, benchmark evaluation, and reproduction of all figures and tables—upon publication. The code will be made available on GitHub with comprehensive instructions to ensure faithful reproduction of results.

Guidelines:

- The answer NA means that paper does not include experiments requiring code.
- Please see the NeurIPS code and data submission guidelines (`https://nips.cc/public/guides/CodeSubmissionPolicy`) for more details.
- While we encourage the release of code and data, we understand that this might not be possible, so "No" is an acceptable answer. Papers cannot be rejected simply for not including code, unless this is central to the contribution (e.g., for a new open-source benchmark).
- The instructions should contain the exact command and environment needed to run to reproduce the results. See the NeurIPS code and data submission guidelines (`https://nips.cc/public/guides/CodeSubmissionPolicy`) for more details.
- The authors should provide instructions on data access and preparation, including how to access the raw data, preprocessed data, intermediate data, and generated data, etc.
- The authors should provide scripts to reproduce all experimental results for the new proposed method and baselines. If only a subset of experiments are reproducible, they should state which ones are omitted from the script and why.
- At submission time, to preserve anonymity, the authors should release anonymized versions (if applicable).
- Providing as much information as possible in supplemental material (appended to the paper) is recommended, but including URLs to data and code is permitted.

6. **Experimental setting/details**

Question: Does the paper specify all the training and test details (e.g., data splits, hyperparameters, how they were chosen, type of optimizer, etc.) necessary to understand the results?

Answer: [Yes]

Justification: The paper clearly specifies test set sizes and evaluation protocols for each benchmark: a 200-problem subset of APPS, the MATH500 subset of MATH, and 108 LiveCodeBench problems from a specified date range to avoid contamination. It also details hyperparameters for DISC (e.g., $\epsilon_0 = 0.15, \theta = 1.0$, temperature $\tau = 0.2$), and discusses how these were selected via ablation studies. Since the work focuses on inference rather than training, training details such as optimizers are not applicable. Pretrained models (e.g., LLaMA-3.1-8B-Instruct) are used directly without further fine-tuning.

Guidelines:

- The answer NA means that the paper does not include experiments.
- The experimental setting should be presented in the core of the paper to a level of detail that is necessary to appreciate the results and make sense of them.
- The full details can be provided either with the code, in appendix, or as supplemental material.

7. **Experiment statistical significance**

Question: Does the paper report error bars suitably and correctly defined or other appropriate information about the statistical significance of the experiments?

Answer: [Yes]

Justification: The paper follows standard field practice by providing statistical variation where appropriate. These conventions are consistent with accepted norms in the LLM inference scaling literature.

Guidelines:

- The answer NA means that the paper does not include experiments.

- The authors should answer "Yes" if the results are accompanied by error bars, confidence intervals, or statistical significance tests, at least for the experiments that support the main claims of the paper.
- The factors of variability that the error bars are capturing should be clearly stated (for example, train/test split, initialization, random drawing of some parameter, or overall run with given experimental conditions).
- The method for calculating the error bars should be explained (closed form formula, call to a library function, bootstrap, etc.)
- The assumptions made should be given (e.g., Normally distributed errors).
- It should be clear whether the error bar is the standard deviation or the standard error of the mean.
- It is OK to report 1-sigma error bars, but one should state it. The authors should preferably report a 2-sigma error bar than state that they have a 96% CI, if the hypothesis of Normality of errors is not verified.
- For asymmetric distributions, the authors should be careful not to show in tables or figures symmetric error bars that would yield results that are out of range (e.g. negative error rates).
- If error bars are reported in tables or plots, The authors should explain in the text how they were calculated and reference the corresponding figures or tables in the text.

8. **Experiments compute resources**

Question: For each experiment, does the paper provide sufficient information on the computer resources (type of compute workers, memory, time of execution) needed to reproduce the experiments?

Answer: [Yes]

Justification: The paper includes a dedicated section in the appendix detailing the compute resources used. Evaluations involving OpenAI proprietary models (e.g., GPT-4o-mini, GPT-3.5) were conducted via the OpenAI API from standard desktop machines with CPU-only inference. For open-source models such as LLaMA-3.1-8B-Instruct, Mistral, and Qwen, inference was run on a single NVIDIA A100 GPU with 80GB of memory. This setup demonstrates that DISC is lightweight and can be reproduced under modest compute conditions. The disclosure is sufficient to support replication of experimental results.

Guidelines:

- The answer NA means that the paper does not include experiments.
- The paper should indicate the type of compute workers CPU or GPU, internal cluster, or cloud provider, including relevant memory and storage.
- The paper should provide the amount of compute required for each of the individual experimental runs as well as estimate the total compute.
- The paper should disclose whether the full research project required more compute than the experiments reported in the paper (e.g., preliminary or failed experiments that didn't make it into the paper).

9. **Code of ethics**

Question: Does the research conducted in the paper conform, in every respect, with the NeurIPS Code of Ethics https://neurips.cc/public/EthicsGuidelines?

Answer: [Yes]

Justification: The paper adheres to the NeurIPS Code of Ethics by utilizing publicly available datasets (APPS, MATH500, LiveCodeBench) that do not contain personally identifiable information, thereby respecting privacy and consent considerations. The research does not involve human subjects, surveillance, or any form of deceptive practices. The proposed DISC method aims to enhance the efficiency of large language model inference, potentially reducing computational resources and associated environmental impact. The paper also includes discussions on limitations and potential biases, aligning with the ethical standards outlined by NeurIPS.

Guidelines:

- The answer NA means that the authors have not reviewed the NeurIPS Code of Ethics.
- If the authors answer No, they should explain the special circumstances that require a deviation from the Code of Ethics.
- The authors should make sure to preserve anonymity (e.g., if there is a special consideration due to laws or regulations in their jurisdiction).

10. **Broader impacts**

Question: Does the paper discuss both potential positive societal impacts and negative societal impacts of the work performed?

Answer: [Yes]

Justification: The paper includes a dedicated Impact Statement in the appendix that discusses both positive and negative societal impacts. On the positive side, DISC improves inference efficiency for LLMs without additional training or hardware, lowering costs and environmental impact while enhancing accessibility. It also promotes interpretability and aligns well with open-source use cases. On the negative side, the paper acknowledges that increased efficiency could accelerate deployment of LLMs in settings with insufficient ethical oversight, potentially contributing to misuse such as misinformation or academic dishonesty. These trade-offs are explicitly addressed in alignment with responsible AI practices.

Guidelines:

- The answer NA means that there is no societal impact of the work performed.
- If the authors answer NA or No, they should explain why their work has no societal impact or why the paper does not address societal impact.
- Examples of negative societal impacts include potential malicious or unintended uses (e.g., disinformation, generating fake profiles, surveillance), fairness considerations (e.g., deployment of technologies that could make decisions that unfairly impact specific groups), privacy considerations, and security considerations.
- The conference expects that many papers will be foundational research and not tied to particular applications, let alone deployments. However, if there is a direct path to any negative applications, the authors should point it out. For example, it is legitimate to point out that an improvement in the quality of generative models could be used to generate deepfakes for disinformation. On the other hand, it is not needed to point out that a generic algorithm for optimizing neural networks could enable people to train models that generate Deepfakes faster.
- The authors should consider possible harms that could arise when the technology is being used as intended and functioning correctly, harms that could arise when the technology is being used as intended but gives incorrect results, and harms following from (intentional or unintentional) misuse of the technology.
- If there are negative societal impacts, the authors could also discuss possible mitigation strategies (e.g., gated release of models, providing defenses in addition to attacks, mechanisms for monitoring misuse, mechanisms to monitor how a system learns from feedback over time, improving the efficiency and accessibility of ML).

11. **Safeguards**

Question: Does the paper describe safeguards that have been put in place for responsible release of data or models that have a high risk for misuse (e.g., pretrained language models, image generators, or scraped datasets)?

Answer: [NA]

Justification: The paper does not release any new pretrained models, image generators, or scraped datasets that would pose a high risk of misuse. The work builds upon existing, publicly available models and datasets, and focuses solely on inference-time methods. Therefore, this question is not applicable.

Guidelines:

- The answer NA means that the paper poses no such risks.

- Released models that have a high risk for misuse or dual-use should be released with necessary safeguards to allow for controlled use of the model, for example by requiring that users adhere to usage guidelines or restrictions to access the model or implementing safety filters.
- Datasets that have been scraped from the Internet could pose safety risks. The authors should describe how they avoided releasing unsafe images.
- We recognize that providing effective safeguards is challenging, and many papers do not require this, but we encourage authors to take this into account and make a best faith effort.

12. **Licenses for existing assets**

Question: Are the creators or original owners of assets (e.g., code, data, models), used in the paper, properly credited and are the license and terms of use explicitly mentioned and properly respected?

Answer: [Yes]

Justification: The paper uses publicly available datasets (APPS, MATH, LiveCodeBench) and models (e.g., GPT-4o-mini via OpenAI API, LLaMA-3.1-8B-Instruct, Mistral, Qwen) and appropriately cites their original sources. These assets were used in accordance with their respective licenses and terms of use. The paper does not redistribute any of these assets and respects the usage guidelines set by their creators.

Guidelines:

- The answer NA means that the paper does not use existing assets.
- The authors should cite the original paper that produced the code package or dataset.
- The authors should state which version of the asset is used and, if possible, include a URL.
- The name of the license (e.g., CC-BY 4.0) should be included for each asset.
- For scraped data from a particular source (e.g., website), the copyright and terms of service of that source should be provided.
- If assets are released, the license, copyright information, and terms of use in the package should be provided. For popular datasets, `paperswithcode.com/datasets` has curated licenses for some datasets. Their licensing guide can help determine the license of a dataset.
- For existing datasets that are re-packaged, both the original license and the license of the derived asset (if it has changed) should be provided.
- If this information is not available online, the authors are encouraged to reach out to the asset's creators.

13. **New assets**

Question: Are new assets introduced in the paper well documented and is the documentation provided alongside the assets?

Answer: [NA]

Justification: The paper does not introduce new assets such as datasets, pretrained models, or novel codebases at submission time. The proposed method, DISC, is described in detail in the paper and appendix, and the authors commit to releasing the code with full documentation upon publication.

Guidelines:

- The answer NA means that the paper does not release new assets.
- Researchers should communicate the details of the dataset/code/model as part of their submissions via structured templates. This includes details about training, license, limitations, etc.
- The paper should discuss whether and how consent was obtained from people whose asset is used.
- At submission time, remember to anonymize your assets (if applicable). You can either create an anonymized URL or include an anonymized zip file.

14. **Crowdsourcing and research with human subjects**

Question: For crowdsourcing experiments and research with human subjects, does the paper include the full text of instructions given to participants and screenshots, if applicable, as well as details about compensation (if any)?

Answer: [NA]

Justification: The research does not involve any crowdsourcing experiments or human subjects. All results are derived from automated evaluation of LLM-generated outputs using standard benchmarks and computational metrics.

Guidelines:

- The answer NA means that the paper does not involve crowdsourcing nor research with human subjects.
- Including this information in the supplemental material is fine, but if the main contribution of the paper involves human subjects, then as much detail as possible should be included in the main paper.
- According to the NeurIPS Code of Ethics, workers involved in data collection, curation, or other labor should be paid at least the minimum wage in the country of the data collector.

15. **Institutional review board (IRB) approvals or equivalent for research with human subjects**

Question: Does the paper describe potential risks incurred by study participants, whether such risks were disclosed to the subjects, and whether Institutional Review Board (IRB) approvals (or an equivalent approval/review based on the requirements of your country or institution) were obtained?

Answer: [NA]

Justification: The study does not involve human participants or any form of user study. Therefore, there were no risks to disclose, and IRB approval was not required.

Guidelines:

- The answer NA means that the paper does not involve crowdsourcing nor research with human subjects.
- Depending on the country in which research is conducted, IRB approval (or equivalent) may be required for any human subjects research. If you obtained IRB approval, you should clearly state this in the paper.
- We recognize that the procedures for this may vary significantly between institutions and locations, and we expect authors to adhere to the NeurIPS Code of Ethics and the guidelines for their institution.
- For initial submissions, do not include any information that would break anonymity (if applicable), such as the institution conducting the review.

16. **Declaration of LLM usage**

Question: Does the paper describe the usage of LLMs if it is an important, original, or non-standard component of the core methods in this research?

Answer: [Yes]

Justification: The paper centrally focuses on improving inference-time performance for LLMs through a novel method called Dynamic Decomposition (DISC). It clearly describes how LLMs are used in both the methodology and experiments, including the use of autoregressive LLMs like GPT-4o-mini and LLaMA-3.1-8B-Instruct. DISC interacts with these models by dynamically querying continuations from intermediate prefixes, making the LLM usage both critical and fully documented throughout the paper.

Guidelines:

- The answer NA means that the core method development in this research does not involve LLMs as any important, original, or non-standard components.
- Please refer to our LLM policy (https://neurips.cc/Conferences/2025/LLM) for what should or should not be described.

