# OpenReview forum: "DISC: Dynamic Decomposition Improves LLM Inference Scaling"
_NeurIPS.cc/2025/Conference — NeurIPS 2025 poster_

### Official Review · Reviewer_PSiF · 2025-07-01

**Clarity:** 3
**Significance:** 3
**Originality:** 3
**Rating:** 5
**Confidence:** 3

**Summary:**

This paper presents a novel test-time inference strategy called DISC, which adaptively decomposes the generation process of large language models (LLMs) by dynamically adjusting the step granularity. Instead of relying on static token-level or sentence-level generation, DISC estimates the “difficulty” of each step and decides whether to accept or further expand it using a z-score-based criterion. The method is model-agnostic, requires no additional training, and is compatible with various search paradigms such as greedy decoding, beam search, and MCTS. Experiments across multiple reasoning benchmarks (APPS, MATH, LiveCodeBench) demonstrate notable performance gains under constrained compute budgets.

**Questions:**

Please refer to Weaknesses.

**Ethical Concerns:**

["NO or VERY MINOR ethics concerns only"]

**Limitations:**

Yes

**Quality:**

3

**Strengths And Weaknesses:**

Strengths:
1. The paper is motivated by a practical problem and is clearly written, making it easy for readers to follow.
DISC is easy to plug into existing inference pipelines, and doesn’t require training or model changes.
2. Solid evaluation across tasks and models, and good ablations around its design choices (e.g., temperature, thresholds).

Weaknesses:
1. A number of recent papers (e.g., DynScaling [1], RINS [2]) also explore ways to adapt compute usage during inference. They differ in method—e.g., using multi-armed bandits, recursive refinement, or heuristic difficulty estimates—but share a similar motivation. The paper doesn’t really discuss these. It would help to clearly highlight what DISC does differently (especially around the level of granularity or how compute decisions are made), and what the practical gains are.
2. The current z-score–based decomposition decision appears to be heuristic. Is there any theoretical justification or learned variant that could better adapt to different task domains or model behaviors?
3. While DISC is compatible with search methods like beam or MCTS, the interaction mechanisms are not well explained. Does DISC affect search efficiency, or introduce redundancy in rollouts?
[1] Wang F, Wan X, Sun R, et al. DynScaling: Efficient Verifier-free Inference Scaling via Dynamic and Integrated Sampling[J]. arXiv preprint arXiv:2506.16043, 2025.
[2] Alabdulmohsin I, Zhai X. Harnessing Language's Fractal Geometry with Recursive Inference Scaling[J]. arXiv preprint arXiv:2502.07503, 2025.

---

> ### Author Rebuttal · Authors · 2025-07-31
>
> We thank the reviewer for their thoughtful and constructive feedback. As NeurIPS does not permit the inclusion of PDFs or anonymous external links in the rebuttal this year, we apologize in advance that we are unable to provide annotated figures or visual clarifications. We have done our best to clearly describe figure content and visual changes. We hope our written explanations are sufficiently clear, and we are happy to incorporate any additional clarifications into the camera-ready version if the paper is accepted.
>
>
> > A number of recent papers (e.g., DynScaling [1], RINS [2]) also explore ways to adapt compute usage during inference. They differ in method—e.g., using multi-armed bandits, recursive refinement, or heuristic difficulty estimates—but share a similar motivation. The paper doesn’t really discuss these. It would help to clearly highlight what DISC does differently (especially around the level of granularity or how compute decisions are made), and what the practical gains are.
>
> We thank the reviewer for this excellent suggestion. We will include a dedicated discussion in **Section 5 (Related Work)** to highlight how DISC differs from recent adaptive compute methods like DynScaling, RINS, and MetaScale. These works indeed share a similar motivation—efficiently allocating inference-time compute—but differ in *what granularity* they operate on, *how compute decisions are made*, and *where adaptation occurs*.
>
> * **DISC (ours)** performs *intra-query*, *span-level* dynamic decomposition. It adaptively partitions a single solution into reasoning steps of varying lengths, refining only where needed using a local, binary refine/stop decision. This allows DISC to focus compute on hard reasoning regions while skipping trivial ones. DISC is model-agnostic, inference-time only, and integrates directly with search algorithms such as greedy, beam, and MCTS.
>
> * **DynScaling** (Wang et al., 2025) operates at the *inter-query* level and uses a multi-armed bandit to allocate more samples to harder queries. Within a single query, it leverages an integrated sampling policy (parallel then sequential) to manage sample reuse, but does not dynamically adjust step sizes or decomposition within responses. Thus, DynScaling adapts compute *across queries*, while DISC does so *within a single query* at the granularity of solution spans.
>
> * **RINS** (Alabdulmohsin & Zhai, 2025) introduces *recursive inference*, repeatedly applying the model itself to refine outputs—akin to reprocessing the entire generation in a self-similar manner. It adapts compute by repeating inference passes globally, rather than targeting specific spans or prefixes. In contrast, DISC adaptively selects which local spans to refine, avoiding redundant work on already confident segments.
>
> * **MetaScale** (Liu et al., 2025) performs *strategy-level adaptation* within a query. It maintains a pool of diverse "meta-thought" prompts or personas and uses UCB-style sampling to pick among them based on a reward model. MetaScale adapts *how* to reason (via prompt strategy) rather than *where* to focus compute in a given reasoning chain.
>
> **Practical implications:** These differences imply complementary strengths. DISC is well-suited for tasks requiring fine-grained correction (e.g., mathematical reasoning or code synthesis), while DynScaling is beneficial for triaging a large batch of mixed-difficulty queries. RINS can improve outputs through recursive refinement, and MetaScale enhances diversity through strategic prompting. DISC's lightweight, *plug-and-play* design enables compatibility with a variety of search methods (Sec. 3.4) and its gains stem from directly identifying critical reasoning junctures (Sec. 3.5). We believe combining these techniques could be a promising future direction, and we appreciate the reviewer’s prompt for this comparison.
>
> > The current z-score–based decomposition decision appears to be heuristic. Is there any theoretical justification or learned variant that could better adapt to different task domains or model behaviors?
>
> Thank you for this perceptive question. Yes, there is a theoretical justification for using the z-score as the decision metric in DISC. As detailed in Section 3.3 of the main paper and Section F of the supplementary material, our optimality analysis relies critically on z-score–based comparisons to guide decomposition decisions. Note that **we updated the Appendix as part of the supplementary material after the full paper submission to strengthen our theoretical justification**—specifically, by adding Lemma 1 and clarifying its role in the proof of Theorem 1—to formally show that z-score–based acceptance ensures monotonic improvement in the likelihood of discovering optimal completions.
>
> Specifically, we assume that rollout rewards approximately follow a location-scale family distribution (e.g., Gaussian or sub-Gaussian)—an assumption we empirically support in Appendix C.5. Under this assumption, the z-score of the maximum sampled reward provides a standardized estimate of how promising a given prefix is, in terms of the tail probability of achieving higher-reward completions. Selecting the prefix with the lower z-score thus corresponds to choosing the one with the highest estimated probability of improvement over the current base prefix.
>
> This acceptance rule plays a central role in our optimality guarantee (Theorem 1, Section 3.3). The key step in the proof is Lemma 1 (Section F of supplementary material), which shows that accepting a candidate prefix only when its z-score is lower than that of the current base ensures that the probability of sampling the optimal suffix does not decrease. In other words, each accepted candidate maintains or improves the likelihood of discovering an optimal solution. This monotonicity is essential for proving that DISC will, under mild assumptions on the LLM policy (i.e., that the optimal solution lies in the support of the model), eventually commit to the optimal solution and terminate successfully, with faster convergence than BoN in expectation. Without this property, the algorithm could regress to less promising regions of the search space and fail to converge.
>
> Furthermore, our ablation study (Figure 8, middle) shows that replacing the z-score with alternative acceptance metrics—such as mean reward (DISC-Q), random selection (DISC-R), or even reversed z-score (DISC-negZ)—substantially degrades performance. Notably, DISC-Z (our method) outperforms all alternatives, while DISC-negZ underperforms even random selection, underscoring the importance of using z-score as the acceptance criterion.
>
>
> > While DISC is compatible with search methods like beam or MCTS, the interaction mechanisms are not well explained. Does DISC affect search efficiency, or introduce redundancy in rollouts?
>
> Thank you for this insightful question. We address your concerns below.
>
> First, we clarify that DISC is designed to be *plug-and-play* with a wide range of search algorithms—including greedy, beam, and Monte Carlo Tree Search (MCTS)—by modularly handling the **decomposition logic** that determines *how far* to extend a node’s prefix during search. Rather than altering the core mechanics of the search algorithm (such as beam ranking, MCTS selection, or backpropagation), DISC acts as a drop-in replacement for the static rule that typically governs when to branch (e.g., after a token or a line). During the node expansion phase—regardless of the specific search method—DISC proposes a candidate step by appending a fraction of the best sampled suffix to the current prefix, and evaluates whether to accept or contract this step using rollout statistics (e.g., z-score comparisons). Because DISC only modifies *when* a node is expanded (i.e., step size) and *not* how nodes are selected or evaluated, it can be seamlessly incorporated into existing search pipelines without changes to the core search logic. This separation of concerns ensures that DISC enhances search granularity and adaptivity without introducing architectural complexity or engineering overhead.
>
> This dynamic step sizing significantly improves *search efficiency* by focusing compute on challenging regions of the reasoning trace. For example, instead of expanding the tree one token at a time (token-level) or by fixed segments (e.g., line-level), DISC adapts step sizes based on rollout statistics, effectively skipping trivial segments while zooming in on uncertain or error-prone areas. This accelerates the search without sacrificing fidelity.
>
> Importantly, DISC does **not introduce redundancy in rollouts**. Rather, it is designed to *reuse the rollout data already collected by the underlying search algorithm*—such as the completions sampled for evaluating candidate nodes in beam search or MCTS. These same completions are jointly used by DISC to compute statistics (e.g., z-scores) for determining whether to accept a proposed step size or to contract it. In this way, DISC leverages the model outputs already required for node evaluation, avoiding duplicated sampling or unnecessary recomputation. This shared use of rollout data ensures that DISC remains efficient and tightly integrated with the existing search pipeline. We have clarified this point further in Appendix E.1.
>
> We will update Appendix E.1 to more explicitly describe how DISC integrates with beam search and MCTS, and thank the reviewer's feedback.

---

### Official Review · Reviewer_HBjZ · 2025-07-01

**Clarity:** 2
**Significance:** 3
**Originality:** 3
**Rating:** 4
**Confidence:** 3

**Summary:**

The paper presents a novel framework for dynamic decomposition in modern reasoning or code generation LLMs, named DISC. DISC extends search strategies like MTCS and beam search by adaptively partitioning solution steps to put more computational resources on harder parts, thus improving effectiveness of inference. The authors performed experiments on APPS, MATH, and LiveCodeBench benchmarks against several static baselines, including token-level, sentence-level, and single-step decomposition. achieving up to 10% reduction in error relative to the baselines.

**Questions:**

- Figure 6a present open-source models. It is unclear which version of the model is base (if any) and which one is DISC (if any). Please clarify.

**Ethical Concerns:**

["NO or VERY MINOR ethics concerns only"]

**Final Justification:**

The discussion clarified some concerns. Keeping the original score. Ok if paper accepted.

**Limitations:**

The paper should contain a clearly marked list of limitations.

**Quality:**

3

**Strengths And Weaknesses:**

Strengths:

* The paper rests on a simple yet effective idea to change the problem decomposition steps based on the insight that prefix statistics (which correlate with the promising reward improvement) can aid the split, leading to larger or smaller steps. The paper also shows that the DISC technique integrates with the state-of-the art LLMs, including open source and commercial ones.

* Evaluation shows reasonable improvements over various baselines, given pass@k and pass@token metrics. Overall, I found the results in this section convincing, suggesting that DISC's adaptive partitioning improves correctness with the same token/sample budget over the alternative simple baselines. While the improvements are not enormous, they are significant enough to merit publication of this work and the overhead of the technique is low (assuming an appropriate reward model exists).

* Several ablation studies show the impact of several parameters on the quality of the search, strengthening the main results. The authors further support their arguments with an extensive appendix with additional experimental studies and mathematical derivations.


Weaknesses:

* The paper does not explicitly discuss the limitations of the proposed approach. For instance, the presentation of how self-generated test validation proceeds is currently terse and the details should be clarified both in the evaluation section and discussed as a limitation.

* (minor) Algorithm: In Algorithm 1, Y_b is a set whose elements are pairs of elements. However the next line is applying the functions max and argmax (presumably on 2nd and 1st element, respectively). Update the presentation so that the computation is mathematically well-defined.

* (minor) Formatting: try to keep the colors consistent. For instance, Fig 7 has different colors for DISC across different benchmarks. Figure 9 would be more understandable if it had means with standard deviations as bars. Also, the figure caption should explain the dotted lines on that plot.

---

> ### Author Rebuttal · Authors · 2025-07-31
>
> We thank the reviewer for their thoughtful and constructive feedback. As NeurIPS does not permit the inclusion of PDFs or anonymous external links in the rebuttal this year, we apologize in advance that we are unable to provide annotated figures or visual clarifications. We have done our best to clearly describe figure content and visual changes. We hope our written explanations are sufficiently clear, and we are happy to incorporate any additional clarifications into the camera-ready version if the paper is accepted.
>
>
> > The paper does not explicitly discuss the limitations of the proposed approach. For instance, the presentation of how self-generated test validation proceeds is currently terse and the details should be clarified both in the evaluation section and discussed as a limitation.
>
> We appreciate the reviewer’s thoughtful feedback and agree that an explicit discussion and marked list of limitations, particularly around self-generated test validation, is important. Per the reviewer's suggestion, we will (1) add a dedicated limitations section, and (2) expand our explanation of the self-generated test evaluation setting in Section 4.1 and Appendix D.3.
>
> Specifically, we now clarify that self-generated test validation refers to a setup where the LLM is prompted to produce its own unit tests for a given code generation problem, and these tests are then used as a proxy reward model for inference-time evaluation. This approach has gained traction due to the high cost or unavailability of ground-truth test cases in many real-world tasks.
>
> There are several interesting limitations of our framework.
>
> * **Reliance on LLM Test Generation Abilities**: The effectiveness of this evaluation depends critically on the LLM’s ability to produce meaningful and comprehensive unit tests. If the generated tests fail to cover edge cases or are semantically shallow, the evaluation may overestimate the correctness of completions.
>
> * **Assumption of Test Set Precision**: The framework assumes that the majority of self-generated tests are themselves correct and executable. In practice, this may not always hold—especially for complex or ambiguous problem specifications—potentially leading to false positives or unreliable reward signals.
>
> * **Comparison to Ground Truth Is Not Guaranteed**: Unlike evaluation with a ground-truth verifier, self-generated tests are not guaranteed to align with the reference solution, which can introduce inconsistencies in scoring across different methods.
>
> These challenges are documented in prior work, including \[1], which systematically studies the tradeoffs and failure cases of using generated tests for code evaluation. We now refer readers to \[1] and related literature for a more comprehensive discussion of the strengths and limitations of this approach.
>
> We also discuss additional limitations in the newly added section, including DISC’s reliance on a reward model, its current design for single-turn generation tasks, and scenarios where early reasoning does not meaningfully influence the final answer.
>
> We hope this clarification and expansion address the reviewer’s concern, and we thank them again for helping us improve the clarity and completeness of the paper.
>
> **Reference**
> \[1] Chen, Bei, et al. *"Codet: Code generation with generated tests."* arXiv preprint arXiv:2207.10397 (2022).
>
> > (minor) Algorithm: In Algorithm 1, Y_b is a set whose elements are pairs of elements. However the next line is applying the functions max and argmax (presumably on 2nd and 1st element, respectively). Update the presentation so that the computation is mathematically well-defined.
>
> > (minor) Formatting: try to keep the colors consistent. For instance, Fig 7 has different colors for DISC across different benchmarks. Figure 9 would be more understandable if it had means with standard deviations as bars. Also, the figure caption should explain the dotted lines on that plot.
>
> We thank the reviewer for their helpful suggestions regarding the algorithm presentation and figure formatting.
>
> To address the first point, we have revised the notation in Algorithm 1 to ensure the computation is mathematically well-defined. Specifically, we now explicitly define the elements of the set $Y_b = \{ (s_i, R(s_i)) \}$ as pairs of completions and their associated rewards. We also clarified that we use $\max_{(s, R(s)) \in Y_b} R(s)$ and $\arg\max_{(s, R(s)) \in Y_b} R(s)$ to compute the maximum reward and the corresponding completion, respectively, for cleaner and unambiguous mathematical notation.
>
> For the second point regarding formatting, we have made several updates for clarity and consistency. We standardized the color palette across figures so that DISC is represented with a consistent color throughout all benchmark plots, including Figure 7. In Figure 9, we have updated the plots to make the trends and variability clearer. Additionally, we expanded the figure caption to explain the meaning of the dotted lines, which indicate baselines or comparative trends depending on the subplot.
>
> We appreciate the reviewer’s careful attention to detail and believe these changes improve the clarity and presentation of the manuscript.
>
> > Figure 6a present open-source models. It is unclear which version of the model is base (if any) and which one is DISC (if any). Please clarify.
>
> We thank the reviewer for their thoughtful feedback and apologize for any lack of clarity in Figure 6a. The left panel of Figure 6 displays the inference scaling performance of DISC applied to three different open-source base models at varying budgets: Llama-3.1-8B, Mistral-7B-v0.3, and Qwen-2.5-7B. Each line color corresponds to a specific model, and we have revised the figure caption and legend in the updated manuscript to make this distinction explicit. As shown in the figure, DISC consistently improves performance across all three models. Notably, while the *relative* accuracy improvements are more substantial for weaker models such as LLaMA (from 0.01 to 0.04; a 300% relative increase) and Mistral (from 0.0 to 0.02), the *absolute* gain is highest for the stronger Qwen model (from 0.095 to 0.17; a 7.5 percentage point increase), as detailed in Section 4.1. These results underscore DISC’s robustness and general applicability, yielding meaningful gains even under limited compute budgets. We thank the reviewer again for bringing this to our attention and have clarified the figure and surrounding text accordingly.

---

> > ### Author Response · Authors · 2025-08-05
> >
> > Hi Reviewer HBjZ,
> >
> > Thank you again for your thoughtful and detailed review. We’ve submitted a rebuttal that addresses your comments, including a more explicit discussion of the limitations—particularly around self-generated test validation—and revisions to Algorithm 1 and figure formatting as you suggested. We also clarified Figure 6a and expanded captions to improve interpretability.
> >
> > If you have a chance to take a look and share any follow-up thoughts, it would be greatly appreciated! Thank you again for your time and contribution to the review process.

---

> > ### Comment · Reviewer_HBjZ · 2025-08-05
> >
> > Thank you for your response. They clarify the main questions that I have. Please include the discussion of limitations in the final version. I will keep my current rating.

---

> > > ### Author Response · Authors · 2025-08-07
> > >
> > > Thank you for your follow-up and your continued positive outlook on the work. We are glad that our response clarified your main questions. As you suggested, we will include a discussion of the limitations in the final version. Your comments have been valuable in helping us strengthen the paper.

---

### Official Review · Reviewer_1o4h · 2025-07-02

**Clarity:** 3
**Significance:** 2
**Originality:** 3
**Rating:** 4
**Confidence:** 3

**Summary:**

The paper addresses a timely and important challenge in large language model (LLM) inference: how to dynamically allocate compute by adaptively decomposing reasoning steps during decoding. The proposed DISC method introduces a dynamic step-size adjustment algorithm guided by statistical reward estimations (e.g., z-scores).

**Questions:**

1.	Include explicit compute-efficiency metrics
2.	Provide more concrete visualizations or case studies showing how DISC reallocates compute dynamically in real examples.
3.	More questions can be seen in the Weakness part.

**Ethical Concerns:**

["NO or VERY MINOR ethics concerns only"]

**Final Justification:**

After several-round rebuttal, the authors have mostly addressed my concern. Considering the feedback and the comments from the other reviewers', I would like to upgrade the final rating to BA, and expect the authors to include the entire response in the final version to imporve the overall quality of the content.

**Limitations:**

Yes.

**Quality:**

2

**Strengths And Weaknesses:**

1. The paper proposes a seemingly effective decomposition mechanism that aims to reduce redundant sampling and better allocate compute across reasoning steps.
2. The experimental evaluation is relatively thorough.


## Weakness：
1.	Although the paper claims DISC performs dynamic step selection based on reward distribution statistics, it lacks a principled theoretical justification for this dynamic behavior. It is unclear what precisely governs the dynamic adaptation—apart from the z-score of a small number of sampled candidates.

2.The method depends strongly on hyperparameters such as the initial partition ratio, the reward normalization scheme, and update rules. Therefore, this dynamic decomposition is fundamentally similar to simply pre-setting a static decomposition strategy.

3.A key premise of the paper is to dynamically allocate compute based on the difficulty of different steps in the decomposition. However, the notion of step difficulty is never formally defined, measured, or validated. There is no metric to demonstrate that the method is indeed allocating more compute to harder reasoning steps.

4.The method is claimed to improve inference efficiency, however, no concrete analysis of runtime is provided. Figure 5 reports only task accuracy metrics such as Pass@token or Pass@k, which do not isolate computational cost. While the authors vaguely claim negligible runtime overhead (Sec. 4.1 and Appendix Fig. 40), no detailed comparison of actual time or resource usage is shown. This significantly weakens the credibility of the efficiency claim.

5.	The method is only compared against a few relatively simple and static decomposition schemes. The paper omits recent and stronger baselines from the literature on adaptive inference and multi-step reasoning with LLMs.

---

> ### Author Rebuttal · Authors · 2025-07-31
>
> We thank the reviewer for their thoughtful and constructive feedback. As NeurIPS prohibits PDFs and anonymous external links in rebuttals this year, we regret being unable to provide annotated figures. We have instead described visual content through text and hope our explanations are clear.
>
> > The paper lacks a principled theoretical justification for dynamic step selection, and it is unclear what precisely governs the dynamic adaptation.
>
> We clarify the intuition behind how DISC’s dynamic adaptation works with a concrete example, and explain why it is theoretically necessary for guaranteeing optimality.
>
> ### Walkthrough of Adaptation Behavior
>
> Consider the example in Sec. 3.5:
> **“What's the maximum area of a rectangle with a 24 inch perimeter?”**
>
> DISC starts by prompting the model with this question and sampling multiple full solutions (Step 1 in figure 2). It selects the best one (e.g., the full derivation involving variables, equations, and a final numeric answer) and tries to reuse the **first few tokens** of that best solution to form a new “candidate prefix.”
>
> Initially, it proposes a candidate prefix like:
>
> > **“Let the length of the rectangle be l and the width of the rectangle be x.”**
>
> DISC samples completions from this prefix and computes its *z-score*, which measures how promising it is (lower is better). In this case, the completions from the candidate prefix were **less helpful** than completions from the original question alone, so DISC **rejects the prefix** (Step 2 in Figure 2). It reduced the step size (i.e., used a shorter snippet) and tries again with a shorter prefix (Step 3 bottom, Figure 2):
>
> > **“Let the length of the rectangle be l”**
>
> This shorter prefix yielded significantly better completions, so DISC **accepted it and committed** to it as the first step (Step 3 top, Figure 2). This **reject-then-contract loop** is key to DISC’s behavior. Each rejection triggers a smaller step size (shorter prefix), and each acceptance locks in a useful piece of reasoning that future completions build on.
>
> This dynamic mechanism also avoids wasting compute on easy steps. For instance, in the *final step*—solving for the area—DISC quickly accepted the candidate prefix, as the model easily produced high-quality completions without needing resampling.
>
> ### Theoretical Justification
>
> Our theoretical analysis (Sec. 3.3, Supplementary material F.1) proves that DISC converges to an optimal solution—faster than BoN, the most widely adopted scaling method. Crucially, this guarantee **requires** comparing prefixes using a criterion like the z-score. The z-score acts as a proxy for whether committing to a prefix increases the likelihood of finding a better solution. If we accepted prefixes arbitrarily or used a non-adaptive rule, this guarantee would not hold.
>
> In short, DISC’s dynamic step adaptation is not just a practical design choice—it is essential for theoretical optimality.
>
> > The method depends on hyperparameters like the initial partition ratio and reward normalization scheme, making it akin to using a fixed decomposition strategy.
>
> We thank the reviewer for this insightful comment. We clarify that DISC introduces only one new hyperparameter—**the initial partition ratio** $\alpha_0$—on top of existing parameters used by search algorithms. This parameter is *not tuned per problem or model*, and is held fixed across all problems and model types (Sec. 4.1, App. C.3), demonstrating robustness and generality.
>
> Importantly, while $\alpha_0$ sets an initial *granularity guess*, it **does not define the decomposition itself**. Instead, DISC launches a **feedback-driven refinement process**, dynamically adjusting decomposition boundaries in response to observed rollout statistics. This stands in contrast to static schemes (e.g., token-level or sentence-level splitting), which impose fixed segmentations regardless of task structure or model behavior.
>
> As illustrated in Section 3.5 and Appendix D.2, DISC can yield *entirely different decompositions for the same problem* depending on which reasoning path the model follows (e.g., solving via Lagrange multipliers vs. first-order derivatives). Static methods, by contrast, produce identical partitions and miss reasoning-path sensitivity. In contrast, DISC adapts segmentation based on model uncertainty—something static methods inherently lack.
>
> Empirically, we show that DISC performs robustly across a wide range of $\alpha_0$ values (0.10–0.30; Fig. 8, right), and improves performance **without any per-task tuning** across diverse models including GPT-4o-mini, LLaMA, Mistral, and DeepSeek-R1.
>
> Finally, **Theorem 1 (Sec. 3.3)** provides a formal guarantee that DISC converges to the optimal solution. In contrast, **static decomposition** can provably fail to reach the optimal solution when used with search algorithms such as greedy or beam search.
>
> **In summary**, DISC is not a static decomposition strategy—it’s a dynamic, feedback-driven process with theoretical backing and strong empirical robustness.
>
> > Step difficulty is undefined, and it's unclear if harder steps receive more compute. Add visualizations or case studies on compute allocation.
>
> We would like to clarify on what we mean by step difficulty:
>
> **1. Defining Step Difficulty.**
> We define a step’s difficulty operationally as the likelihood that additional sampling will yield better completions from that prefix. Under our scale-free reward assumption (Appendix C.5), the z-score of reward samples serves as a proxy: higher z-scores imply lower odds of improvement and thus greater difficulty. This is formalized in Section 3.2 and illustrated in Figure 4.
>
> **2. How DISC Allocates Compute.**
> DISC is explicitly designed to adaptively allocate more sampling to difficult steps via contractive refinement. When a proposed prefix exhibits a high z-score—signaling a challenging step—DISC contracts the step size and samples more finely around that region to explore improved continuations. Conversely, when the z-score is low, indicating an easier step, DISC proceeds with coarser granularity and fewer samples. This recursive refinement mechanism is implemented via a simple rejection step (Algorithm 1, Lines 11–15): if a candidate prefix exhibits a higher z-score than the base prefix, it is rejected and the step size is reduced. As such contractions are more frequent on harder steps, DISC naturally concentrates sampling—and hence compute—on regions with higher z-scores.
>
>
> **3. Empirical Evidence.**
> To address the reviewer’s suggestion for additional concrete evidence:
>
> * We have created a new **scatter plot** showing a **positive correlation** between prefix z-score and number of samples of at that step, directly validating that DISC concentrates effort on harder steps.
> * As a case study, **Section 3.5 and Appendix D.2** present decompositions of real examples, with each step color-coded by z-score. These illustrate that DISC devotes more sampling (i.e., compute) to steps like "which" and "therefore"—short but pivotal reasoning points for LLMs, in line with prior observations on autoregressive generation sensitivity.
> * For example, in the MATH problem shown in Section 3.5, DISC allocated 35 out of 100 calls to the first step and 49 to the third step (both with higher z-scores), while the second step received 12 calls and the final step received only 4 calls, demonstrating targeted compute allocation.
>
>
> > The method omits comparisons against recent baselines in adaptive inference and multi-step reasoning.
>
> Thank you for the suggestion. We have implemented and evaluated two recent adaptive inference baselines—**AdaPrune** (Zhao et al., 2024) and **S1** (Muennighoff, 2025). As shown below, **DISC continues to outperform both under a fixed compute budget of 10,000 output tokens** using GPT-4o-mini on APPS:
>
> | Method   | Pass\@10,000 Tokens |
> | -------- | ------------------- |
> | DISC     | **0.550**               |
> | AdaPrune | 0.500               |
> | S1       | 0.460               |
>
> These results further support the effectiveness of dynamic decomposition for inference scaling.
>
> We also clarify that DISC addresses a complementary goal to prior work: rather than improving *how to search*—via algorithmic design (Wang, 2025), prompt or plan optimization (Feng, 2023), or reward-based inference (Zhang et al., 2024)—DISC introduces a dynamic *decomposition* mechanism that adaptively controls branching granularity and timing. As shown in Section 3.4 and Appendix E, DISC is agnostic to the underlying search policy and works plug-and-play with methods from Zhang et al. (2024), Feng (2023), and Wang (2025).
>
>
> > The absence of explicit runtim timing or resource comparisons weakens the efficiency claim. Include explicit compute-efficiency metrics.
>
> Thank you for highlighting the need for explicit compute-efficiency evaluation. In response, we conducted a new experiment comparing methods under a fixed 3-minute wall-clock budget on APPS using GPT-4o-mini:
>
> | Method     | Pass\@3min |
> | ---------- | ---------- |
> | LineSplit  | 0.530      |
> | TokenSplit | 0.515      |
> | BoN        | 0.525      |
> | DISC       | **0.555**  |
>
> This result supports our claim: **DISC yields better performance within the same runtime constraints**.
>
> We also wish to clarify that LLM inference latency is heavily influenced by environment-specific factors such as batching, caching, hardware, and API response times from the model provider. As discussed in Sec. 4.1 and Appendix D.6 (Fig. 43), the majority of runtime is dominated by LLM generation latency, not the overhead introduced by our method. Because wall-clock time can vary widely across platforms and deployment contexts, token- and query-based metrics—such as Pass@token and Pass@k—are widely adopted in the literature (e.g., (Snell, 2024)) as platform-agnostic and reproducible proxies for compute cost. Moreover, tokens are the standard unit of billing and compute cost used by inference providers.

---

> > ### Author Response · Authors · 2025-08-05
> >
> > Hi Reviewer 1o4h,
> >
> > Thank you again for your review and for raising several important points. We’ve submitted a detailed rebuttal aimed at clarifying key aspects of our method, including a step-by-step walkthrough example illustrating DISC’s dynamic adaptation behavior, the theoretical motivation for adaptive step selection, and how step difficulty governs compute allocation.
> >
> > We’ve also added new comparisons against recent adaptive inference baselines (AdaPrune and S1), clarified how DISC differs from static decomposition approaches, and included a runtime-based evaluation to support our efficiency claims.
> >
> > If you have time to revisit the rebuttal, we’d be very grateful for any further thoughts, and we’d be more than happy to answer or clarify anything else if needed.
> >
> > Thanks again for your time and effort in reviewing our submission.

---

> > ### Comment · Reviewer_1o4h · 2025-08-06
> > **Concerns regarding runtime efficiency**
> >
> > I appreciate the detailed responses from the authors, which have addresses my initial concerns in part.
> >
> > Actually, I still have some remaining concerns regarding runtime efficiency. As the authors themselves note (e.g., in Response 1), the proposed method relies on repeated prompting and z-score computation for each candidate prefix. This naturally raises the question: would such repeated sampling and evaluation not increase inference time? If so, how does the method still qualify as being more efficient in terms of runtime?

---

> ### Author Response · Authors · 2025-08-07
>
> We appreciate the reviewer’s follow-up and interest in the work. Below, we clarify how **inference scaling** is evaluated, explain the meaning of metrics like **Pass\@k**, **Pass\@token**, and **Pass\@Runtime**, and outline why **DISC is more runtime-efficient**—even though it involves repeated sampling and z-score computations.
>
> ---
>
>
> ### What is Inference Scaling?
>
> The goal of inference scaling is to design methods that **make more effective use of available test-time compute**. A method exhibits *better inference scaling* if, when given **more compute**—in the form of more samples, longer generations, or more total inference time—it can convert that additional compute into **improved accuracy** or performance.
>
> In this context, **efficiency** refers to the **accuracy per unit of compute**, not the speed of a single model call. A method is considered more runtime-efficient if it solves more problems correctly within the same compute budget.
>
> ---
>
> ### Clarifying Evaluation Metrics
>
> We evaluate inference scaling using three metrics:
>
> * **Pass\@k**: The probability that at least one out of **k sampled solutions is correct**. For instance, Pass\@10 asks: if the method makes 10 model calls, does any one of them yield the correct answer?
>
> * **Pass\@token**: The probability that the method produces a correct solution **within a fixed token budget**, capturing **token efficiency**.
>
> * **Pass\@Runtime**: The proportion of problems solved **within a fixed wall-clock runtime budget**, reflecting actual **end-to-end efficiency**. However, this is not a standard metric due to the reasons described before such as how wall-clock time can vary widely across platforms and deployment contexts [1-7].
>
> All three metrics assess how **effectively a method uses a given budget to find correct answers**. Importantly, they compare methods at *equal compute cost*.
>
> > **Put differently**: if two methods both use 10 samples or 10,000 tokens, and one finds the correct answer while the other does not, the former is more runtime-efficient in the inference scaling sense.
>
> ---
>
> ### Example inference scaling method: BoN
>
> **Best-of-N (BoN)** is a widely used and strong inference scaling baseline [1,2]. It samples **N full solutions** from the LLM, scores each (e.g., using a verifier or reward model), and returns the best one. As **N increases**, BoN improves, demonstrating classic inference scaling: more compute leads to higher accuracy.
>
> However, BoN allocates compute **uniformly**—all samples are full solutions, and no effort is made to prioritize more informative or uncertain regions of the generation.
>
> ---
>
> ### Why DISC Is More Efficient
>
> DISC improves over BoN by making **adaptive compute allocation decisions**:
>
> * It decomposes solutions into reasoning steps.
> * It identifies **critical or uncertain steps**, based on model feedback (e.g., z-scores).
> * It focuses sampling on those steps while avoiding redundant sampling elsewhere.
>
> This adaptive mechanism enables DISC to **solve more problems with the same number of samples, tokens, or runtime**. For instance:
>
> * Under a fixed Pass\@10 budget, DISC outperforms BoN on all benchmarks (Figure 7).
> * Under a fixed token budget (Pass\@token), DISC achieves **higher accuracy per token**, at almost all token budgets (Figures 5, 6).
> * Under a fixed runtime budget (Pass\@Runtime), as shown in the rebuttal, DISC solves more problems **within the same wall-clock time**, despite performing internal scoring steps.
>
> ---
>
> ### On Runtime Overhead
>
> Although DISC includes z-score computation and branching logic, the **dominant cost in LLM inference is still token generation**. As shown in Appendix D.6, DISC’s runtime overhead is negligible, and the method achieves higher Pass\@Runtime across all settings.
>
> ---
>
> ### Summary
> * **Inference scaling** is about converting more test-time compute into better accuracy—not about reducing latency.
> * **Pass\@k**, **Pass\@token**, and **Pass\@Runtime** all measure efficiency under fixed budgets.
> * **BoN** improves with more samples but applies compute uniformly across full generations.
> * **DISC adapts compute allocation dynamically**, leading to better sample, token, and runtime efficiency.
> * As seen in experiments and the rebuttal, DISC consistently solves more problems under equal budgets than strong baselines.
>
> We hope this answers your questions and concerns. If so, we would appreciate if you could update your evaluation in light of these clarifications and experiments. If not, we are happy to answer any further questions!

---

> > ### Comment · Reviewer_1o4h · 2025-08-07
> >
> > Thanks for the feedback, which has generally addressed my concern. I would like to upgrade my score to BA, and strongly encourage the authors to include the response during finalisation.

---

> > > ### Author Response · Authors · 2025-08-07
> > >
> > > Thank you for your thoughtful follow-up and for upgrading your score. We're glad our response helped address your concerns, and we appreciate your suggestion to include the clarifications in the final version — we will make sure to do so. Thank you again for your time and feedback throughout the review process.

---

### Official Review · Reviewer_WYfN · 2025-07-03

**Clarity:** 3
**Significance:** 3
**Originality:** 3
**Rating:** 4
**Confidence:** 3

**Summary:**

The paper introduces DISC, a dynamic decomposition approach to inference in large language models that adaptively introduces partitions among the solution steps to reflect the estimated difficulty of the solution, instead of using a fixed and hand-designed set of solution steps or uniform steps. The recursive reward-based process used by DISC decides to dedicate more compute to difficult reasoning steps and less to easy ones and fits naturally with standard search algorithms (greedy, beam, MCTS). On various benchmarks of code generation and mathematical reasoning (APPS, MATH, LiveCodeBench), DISC achieves on average large improvements in inference efficiency and correctness compared to static decomposition baseline results.

**Questions:**

Could the authors clarify the 3 points raised in the Weaknesses section?

**Ethical Concerns:**

["NO or VERY MINOR ethics concerns only"]

**Final Justification:**

I lean towards acceptance of the paper. The authors provided sufficient responses to all concerns that I raised.

**Limitations:**

Yes

**Quality:**

3

**Strengths And Weaknesses:**

Strengths:
1. The paper is well-motivated and tackles a meaningful problem in LLM reasoning
2. Extensive experiments on multiple benchmarks with strong improvements
3. Ablation studies are thorough

Weaknesses:
1. The theoretical results are built on strong assumptions that may not necessarily hold in practice. Moreover, insights that DISC will eventually find the optimal solution as the search budget increases are fairly straightforward and hold for any flexible search algorithm.
2. The z-score acceptance criterion needs further justification
3. The paper lacks analyses of any failure modes or empirical limitations of DISC

---

> ### Author Rebuttal · Authors · 2025-07-31
>
> We thank the reviewer for their thoughtful and constructive feedback. As NeurIPS does not permit the inclusion of PDFs or anonymous external links in the rebuttal this year, we apologize in advance that we are unable to provide annotated figures or visual clarifications. We have done our best to clearly describe figure content and visual changes. We hope our written explanations are sufficiently clear, and we are happy to incorporate any additional clarifications into the camera-ready version if the paper is accepted.
>
> > The theoretical results are built on strong assumptions that may not necessarily hold in practice. Moreover, insights that DISC will eventually find the optimal solution as the search budget increases are fairly straightforward and hold for any flexible search algorithm.
> Got it—here’s another take that brings the “DISC *provably* converges faster than BoN” claim forward, tucking it right after we state the goal of the analysis (so readers see the practical payoff before diving into the technical assumptions). Everything else is left essentially intact.
>
> We agree that our theoretical results rely on strong assumptions, and verifying their validity in practice is problem-specific—a nuance we explicitly discuss in the paper \[Section C.5, Supplementary material and line 159]. Problem-specific verification is a necessity in any analysis of large models. Our analysis seeks to identify the *minimal* conditions under which real-time inference with adaptive decomposition remains viable. **Crucially, it also shows that DISC *provably* converges faster than BoN—the most widely adopted inference scaling method—demonstrating tangible gains from dynamic decomposition even in compute-limited settings.** In particular, the analysis motivates Assumption 4 (Reward distribution converges slowly), which was included in the **updated Appendix as part of the Supplementary Materials submission**. This assumption is both natural and practical: it posits that if the candidate prefix does not significantly improve sample quality, then the reward distribution must remain sufficiently broad to avoid prematurely collapsing exploration. In other words, the algorithm should not commit to a sharp but unreliable prefix unless there is strong evidence of progress. This condition aligns with how LLMs behave in practice—early prefixes often exhibit noisy or uncertain completions, and overly narrow sampling can stunt further search. Assumption 4 effectively encodes a safeguard against overconfident commitment and can even be implemented as an **explicit acceptance criterion**, as described in Appendix F.3: a candidate prefix could be accepted only if both the reward improvement is significant *and* the distributional variance has not collapsed. Although we do not enforce this explicitly in Algorithm 1 for simplicity, doing so would only strengthen DISC’s robustness guarantees.
>
> Moreover, DISC’s convergence is categorically different from that of conventional search algorithms because DISC is not, in itself, a search policy—it is a *representation refinement process*. Rather than deciding *which* branches to explore (as in classical search), DISC determines *how far* to extend the current branch before committing to further sampling, with any sampling strategy one wishes. This design enables DISC to defer commitment until confidence is higher, which in turn reduces the likelihood of prematurely discarding high-reward completions—an issue that plagues greedy and beam search under static decomposition.
>
> Importantly, many commonly used inference strategies—such as greedy search and beam search—**do not guarantee convergence** to the optimal solution when paired with static decomposition. For instance, both can prematurely commit to suboptimal prefixes under static decomposition, irreversibly pruning paths that would have led to higher-reward completions. DISC avoids this pitfall by dynamically refining step sizes based on rollout statistics, allowing it to course-correct and focus compute on promising regions. Our theoretical analysis helps formalize this behavior and provides a conceptual explanation for DISC’s observed monotonic improvement (Sec. 3.3). Furthermore, we empirically validate key assumptions used in our analysis, such as the location-scale nature of reward distributions (App. C.5), reinforcing the practical relevance of our framework.
>
> > The z-score acceptance criterion needs further justification
>
> Thank you for raising this important point. The choice of z-score as the acceptance criterion is theoretically motivated and empirically validated in our work.
>
> From a theoretical standpoint, the z-score quantifies the standardized distance between the maximum reward of a candidate prefix and the mean of its sampled rewards, normalized by their standard deviation. Under the common assumption that rollout rewards approximately follow a location-scale family distribution (e.g., Gaussian or sub-Gaussian)—a property we verify empirically in Appendix C.5—this z-score allows us to estimate the probability that further sampling from a prefix will yield a better solution. Thus, minimizing the z-score corresponds to prioritizing prefixes with a higher estimated probability of improvement.
>
> This criterion is central to our optimality guarantee (Theorem 1 in Section 3.3), which ensures that DISC converges to an optimal solution at least as efficiently as non-decomposed sampling strategies (e.g., BoN), provided that the LLM policy assigns nonzero probability to the optimal completion. The proof relies on Lemma 1 in the supplementary material (Appendix F), which shows that accepting a candidate prefix only when its z-score is lower than that of the base ensures that the probability of sampling the optimal suffix does not decrease. In other words, each accepted prefix refinement either maintains or improves the likelihood of discovering an optimal solution. This monotonicity is key to establishing that the search will not prematurely exclude optimal completions and will eventually reach them given sufficient budget. Without such a decision rule, the algorithm could regress into lower-probability regions of the search space and fail to converge.
>
> Empirically, we further justify this choice through an ablation study (Figure 8, middle), where we compare DISC-Z (our method) to several alternatives such as accepting based on raw mean reward (DISC-Q), random acceptance (DISC-R), and even reversed z-score (DISC-negZ). Among these, DISC-Z performs best, while DISC-negZ underperforms even random selection—underscoring the importance of z-score for effective decomposition.
>
>
>
>
> > The paper lacks analyses of any failure modes or empirical limitations of DISC
>
> We thank the reviewer for the valuable suggestion. In response, we propose the following limitations section to add to the revised manuscript. While DISC demonstrates strong empirical performance and generality across tasks and models, we acknowledge several limitations:
>
> 1. **Dependency on Reward Model Availability**: DISC requires access to a scalar reward model to guide step-wise decomposition. While many tasks such as code generation and math reasoning provide natural verifiers (e.g., test cases or numerical checks), applying DISC in tasks lacking clear outcome signals may require additional engineering, such as constructing LLM-based critics or learned heuristics.
>
> 2. **Most Effective for Single-Turn Generation**: Our current formulation assumes a single-turn generation setting, where a full solution is produced in a single pass. DISC does not directly account for multi-turn dialogue or settings where intermediate steps trigger dynamic interactions with the environment or user. Extending DISC to such settings would require new mechanisms for handling interactive feedback loops.
>
> 3. **Failure to Improve in Trivial or Non-Compositional Tasks**: DISC allocates sampling budget to steps that appear difficult based on prefix statistics. If the task is trivially solvable (e.g., many simple MATH problems) or if early reasoning does not meaningfully constrain the final answer (e.g., hallucinated completions), the benefits of decomposition may be minimal.
>
> We hope this detailed discussion clarifies the scope of DISC and our interest in future extensions to broader task formats and evaluation protocols.

---

> > ### Author Response · Authors · 2025-08-05
> >
> > Hi Reviewer WYfN,
> >
> > Thank you again for your thoughtful and constructive review. We’ve submitted a detailed rebuttal addressing the concerns you raised, including a more explicit discussion of the theoretical assumptions (especially Assumption 4), justification for the z-score acceptance criterion, and an added limitations section as you suggested.
> >
> > If you have a chance to take a look and share any follow-up thoughts, we would greatly appreciate it!
> >
> > Thanks again for your time and effort in reviewing our submission.

---

> > > ### Comment · Reviewer_WYfN · 2025-08-06
> > >
> > > I thank the authors for their detailed response, which largely addresses my concerns. I will maintain my original score and raise my confidence from 2 to 3.

---

> > > > ### Author Response · Authors · 2025-08-07
> > > >
> > > > Thank you for your follow-up and for maintaining a positive outlook on our work. We're grateful that our response helped clarify your concerns, and we appreciate your decision to raise your confidence in the paper. Your feedback has been very helpful in improving both the clarity and presentation of our work.

---

### Note · Authors · 2025-08-15

We would like to sincerely thank all reviewers for their thoughtful feedback, constructive suggestions, and engaging discussion during the rebuttal phase. We greatly appreciate the time and care you devoted to assessing our work.

Across the board, reviewers either maintained a positive rating or increased their score or confidence in the paper. In particular, two reviewers upgraded their evaluations (WYfN: confidence increased from 2->3; 1o4h: score increased to 3->4), while the others reaffirmed their positive assessments and encouraged us to retain clarifications in the final version.

We are encouraged that reviewers highlighted several strengths of the manuscript, including:

* Clear motivation and problem formulation
* A simple yet effective method that is model-agnostic and integrates easily with existing search algorithms
* Solid experimental results with consistent improvements across diverse benchmarks and models
* Thorough ablations and theoretical grounding, with empirical validation of key assumptions

We will incorporate all suggested improvements in the final version, including:

* Adding a dedicated, clearly marked limitations section (as suggested by WYfN and HBjZ)
* Expanding our discussion of related work to contrast DISC with recent adaptive inference approaches such as DynScaling, RINS, and MetaScale (per PSiF’s suggestion)
* Retaining and integrating the additional clarifications, and runtime-constrained evaluations introduced during the rebuttal phase
* Revising Algorithm 1 notation, standardizing figure colors and captions, and clarifying self-generated test validation details

We are grateful for the constructive dialogue and believe these refinements will further strengthen both the clarity and technical contribution of the paper. Thank you again for helping us improve this work.

---

### Decision · Program_Chairs · 2025-09-17

**Decision:**

Accept (poster)

**Comment:**

The paper introduces DISC, a dynamic decomposition approach for inference in large language models (LLMs). Instead of using fixed, static steps, DISC adaptively partitions the solution process. It uses a reward-based statistical criterion (a z-score) to estimate the "difficulty" of a reasoning step. Based on this, it dynamically decides whether to commit to a generated prefix or to shorten the step and dedicate more computational effort to that challenging part. This method is model-agnostic, requires no extra training, and integrates with standard search algorithms like beam search and MCTS. Experiments on code generation and mathematical reasoning benchmarks show that DISC improves inference efficiency and correctness compared to static decomposition methods.

The introduced approach has a clear motivation and it is simple and effective. The experimental results are strong -- showing consistent improvement over multiple benchmarks. Reviewers originally criticised the work for insufficient justification and analysis of the theoretical assumption, e.g. the choice of the z-score criterion, the definition of "step difficulty," and the lack of runtime efficiency metrics. They also concerned about the missing comparisons with the baseline models. However, these concerns were well addressed by the authors during rebuttal, which led to the final consensus from the reviewers that the paper is acceptable.

I agree with the reviewers final assessment and therefore recommend this paper to be accepted.